# A Tgfbr1/Snai1-dependent developmental module at the core of vertebrate axial elongation

André Dias[1], Anastasiia Lozovska[1†], Filip J Wymeersch[2†‡], Ana Nóvoa[1], Anahi Binagui-Casas[2], Daniel Sobral[1§], Gabriel G Martins[1,3], Valerie Wilson[2], Moises Mallo[1*]

[1]Instituto Gulbenkian de Ciência, Oeiras, Portugal; [2]Centre for Regenerative Medicine, Institute for Regeneration and Repair, University of Edinburgh, Edinburgh, United Kingdom; [3]Faculdade de Ciências da Universidade de Lisboa, Lisboa, Portugal

**\*For correspondence:**
mallo@igc.gulbenkian.pt

[†]These authors contributed equally to this work

**Present address:** [‡]RIKEN Center for Biosystems Dynamics Research, Hyogo, Japan; [§]UCIBIO, Departamento de Ciências da Vida, Faculdade de Ciências e Tecnologia, Universidade NOVA de Lisboa, Caparica, Portugal

**Competing interests:** The authors declare that no competing interests exist.

**Abstract** Formation of the vertebrate postcranial body axis follows two sequential but distinct phases. The first phase generates pre-sacral structures (the so-called primary body) through the activity of the primitive streak on axial progenitors within the epiblast. The embryo then switches to generate the secondary body (post-sacral structures), which depends on axial progenitors in the tail bud. Here we show that the mammalian tail bud is generated through an independent functional developmental module, concurrent but functionally different from that generating the primary body. This module is triggered by convergent Tgfbr1 and Snai1 activities that promote an incomplete epithelial to mesenchymal transition on a subset of epiblast axial progenitors. This EMT is functionally different from that coordinated by the primitive streak, as it does not lead to mesodermal differentiation but brings axial progenitors into a transitory state, keeping their progenitor activity to drive further axial body extension.

## Introduction

Formation of the vertebrate body is a complex and dynamic process involving a series of sequential growth and patterning activities. In amniotes, the primordia of the different organs and body structures are laid down progressively in a head to tail sequence by dedicated axial progenitors with stem cell-like properties (*Stern et al., 2006*). Although continuous, this process can be divided into distinct steps based on structural and regulatory features. At the end of gastrulation, during the so-called primary body formation (i.e. post-occipital region of the head, neck and trunk) axial progenitors are located in the epiblast, an epithelial layer at the caudal embryonic end, and their activity is organized by the primitive streak (PS) (*Aires et al., 2018*; *Holmdahl, 1925*; *Steventon and Martinez Arias, 2017*). At this stage, the embryo contains axial progenitors with different potencies. These include the neuro-mesodermal progenitors (NMPs), a bipotent cell population that can generate both neural and mesodermal tissues, and the lateral and paraxial mesoderm progenitors (LPMPs), with potential limited to mesodermal lineages, that together with the endoderm generate organic systems involved in most vital and reproductive functions (*Wilson et al., 2009*; *Wymeersch et al., 2016*). After PS regression and caudal neuropore closure, the embryo engages in secondary body formation (essentially the tail). At this stage, axial progenitors, which are now restricted to NMPs, are located in the tail bud (*Bénazéraf and Pourquié, 2013*; *Henrique et al., 2015*; *Steventon and Martinez Arias, 2017*; *Wilson et al., 2009*).

A large number of genetic experiments led to the identification of factors regulating axial progenitor activity. Some of these factors, including *Wnt3a*, *Fgf8*, *T(Brachyury)* or the *Cdx* gene family,

are required during both primary and secondary body axis formation, as their partial or total inactivation produce different degrees of axial truncations depending on the levels of gene activity left available to the axial progenitors (*Boulet and Capecchi, 2012*; *Greco et al., 1996*; *Herrmann et al., 1990*; *Naiche et al., 2011*; *Savory et al., 2011*; *Takada et al., 1994*). Other factors show regional specific activity, determining whether progenitors generate trunk or tail structures (*Aires et al., 2019*; *Aires et al., 2018*; *Aires et al., 2016*; *DeVeale et al., 2013*; *Robinton et al., 2019*; *Wymeersch et al., 2019*). Gain and loss of function experiments in the mouse revealed a central role for *Pou5f1* (also known as *Oct4*) in trunk development. Indeed, conditional *Pou5f1* inactivation after it had fulfilled its role during preimplantation and early post-implantation stages resulted in embryos lacking trunk structures but still containing recognizable tails (*DeVeale et al., 2013*). Conversely, sustained transgenic *Pou5f1* expression in the axial progenitor region extended trunk development at the expense of the tail (*Aires et al., 2016*). *Pou5f1* importance for vertebrate trunk development was further revealed by the finding that the remarkably long trunks characteristic of the snake body plan seemed to derive from a chromosomal rearrangement involving the *Pou5f1* locus during vertebrate evolution that placed this gene under the control of regulatory elements that maintained its expression for very long developmental periods (*Aires et al., 2016*).

In the tail bud, axial progenitor activity is independent of *Pou5f1* (*DeVeale et al., 2013*). Genetic experiments in mouse embryos revealed that in this area the *Lin28/let-7* axis together with *Hox13* genes, particularly those belonging to the *HoxB* and *HoxC* clusters, occupy a prevalent position in the regulatory hierarchy of axial progenitors in the tail bud (*Aires et al., 2019*; *Robinton et al., 2019*). Interestingly, while tail bud progenitors show drastic responses to variations in the Lin28/let-7 pathway and are strongly inhibited by premature activation of *Hox13* genes, their trunk counterparts are largely non-responsive to those activities (*Aires et al., 2019*; *Robinton et al., 2019*), suggesting the existence of differences in cell competence, at the progenitor level, during primary and secondary body formation. The differences in progenitor regulation at trunk and tail levels seem to be also associated with changes in their functional characteristics. For instance, while NMPs produce neural tube at both axial levels, they are thought to follow different mechanisms in the two regions (*Catala et al., 1995*; *Schoenwolf, 1984*; *Schoenwolf and Smith, 1990*). Differences can also be observed in the properties of their mesodermal derivatives. In particular, while disconnecting the *Lfng* cycling activity disturbs somitogenesis at trunk levels, it has no or minor effects in the tail (*Shifley et al., 2008*; *Williams et al., 2014*). Conversely, forced *Hoxb6* expression blocks tail somitogenesis but has no effect at trunk levels (*Casaca et al., 2016*).

Despite these major differences in axial progenitor regulation and competence, lineage tracing experiments indicate that post-occipital neural and mesodermal structures are generated from a progenitor pool that is, at least to some extent, maintained along the main vertebrate body axis (*Tzouanacou et al., 2009*), thus implying that transition from primary to secondary body development entails unknown molecular mechanisms occurring at the progenitor level (*Aires et al., 2018*). We have previously shown that *Gdf11* activity plays a relevant role in this process (*Aires et al., 2019*; *Jurberg et al., 2013*). However, partial redundancy by *Gdf8* (*McPherron et al., 2009*) complicates proper evaluation of this process, as trunk-to-tail transition eventually becomes activated in *Gdf11* mutant embryos, although at more caudal axial levels and generating a number of abnormal structures (*Aires et al., 2016*; *McPherron et al., 1999*). Therefore, to evaluate the underlying molecular mechanisms, we compared the molecular characteristics of axial progenitors at progressively later developmental stages using a single-cell RNA-sequencing (scRNA-seq) approach and found that the switch from primary to secondary body development entails an incomplete epithelial to mesenchymal transition (EMT) affecting a subset of axial progenitors in the epiblast. Further analyses showed that this process is functionally different from that generating primary body structures and that it is driven by the sequential activity of *Tgfbr1* and *Snai1*. Together, our data uncovered a distinct functional developmental module generating the tail bud from a subset of epiblast axial progenitors, that initiates secondary body formation in mouse embryos.

## Results

### Axial progenitors undergo an incomplete EMT during axial extension

To evaluate the underlying mechanisms of the changes in progenitor activity as they switch from generating primary to secondary body structures, we compared their molecular characteristics at progressively later developmental stages using a scRNA-seq approach. We first obtained scRNA-seq data from the caudal lateral epiblast (CLE) of early head fold mouse embryos [~embryonic stage (E) 8.0], which contain axial progenitors (*Cambray and Wilson, 2002*; *Tam and Behringer, 1997*; *Figure 1A*). The dissected tissue included some nascent mesoderm, but these cells segregated from those of the epiblast using the single-cell consensus clustering (SC3) framework (*Kiselev et al., 2017*), producing two well-defined clusters (*Figure 1A* and *Figure 1—figure supplement 1A*). We then compared these data with published scRNA-seq data from CLE regions of E8.5 and E9.5 embryos (*Gouti et al., 2017*), which similarly to the E8.0 single-cells, contained some nascent mesoderm. Principal component analysis (PCA) of all these single-cell transcriptomes indicated that the epiblast cluster from E8.0 embryos becomes more similar to the mesodermal component (*Figure 1A*; *Figure 1—figure supplement 1B,C*). Interestingly, differential gene expression analysis revealed an increase of mesenchyme-associated genes and a concomitant decrease of some epithelial markers (*Kalluri and Weinberg, 2009*; *Lamouille et al., 2014*) in epiblast clusters as development proceeds (*Figure 1B*). For instance, we observed a transition from *Cdh1* (*E-cadherin*) to *Cdh2* (*N-cadherin*), as well as downregulation of *Epcam* and up-regulation of *Vim*. This tendency was also observed in cells co-expressing *Sox2* and *T* (*Figure 2*), typically associated with NMPs (*Cambray and Wilson, 2007*; *Koch et al., 2017*; *Martin and Kimelman, 2012*; *Olivera-Martinez et al., 2012*; *Tsakiridis et al., 2014*; *Wymeersch et al., 2016*). These results suggest that progress into tail bud stages requires that axial progenitors undergo an EMT.

We further tested this idea in two additional ways. As the majority of axial progenitors in the Chordoneural Hinge (CNH; the region of the tail bud that drives tail elongation) are descended from cells in the Node-Streak Border (NSB) (*Cambray and Wilson, 2007*), we compared the transcriptome of these two regions of the mouse embryo using data from *Wymeersch et al., 2019*. This analysis revealed increased expression of mesenchymal-related genes (e.g. *Snai1*, *Vim* and *Zeb1*) and downregulation of some epithelial markers (e.g. *Cdh1*, *Krt18* and *Cldn4*) in the CNH (*Figure 1D*). In a complementary approach, we examined the transcriptome of a cell population highly enriched in axial progenitors isolated from E10.5 tail buds (*de Lemos et al., 2019*), using as a reference the transcriptome of unsorted tail bud tissue (*Aires et al., 2019*). Again, many mesenchymal markers were highly expressed in the sorted axial progenitors, at levels comparable to those observed in the total tail bud (*Figure 1E*). Conversely, expression of many epithelial markers was significantly lower in the progenitor pool than in the reference tail bud. Therefore, the gene expression profile of tail bud axial progenitors is more consistent with a mesenchymal than with an epithelial phenotype, further suggesting that tail bud formation entails an EMT on the axial progenitors. Interestingly, however, in all our analyses the loss of epithelial characteristics in tail bud axial progenitors was not complete. Indeed, some epithelial markers maintained high expression levels (e.g. *Tjp1*, *Lamb1*) or were upregulated (e.g. *Krt10* in the CNH) when axial progenitors moved into the tail bud (*Figure 1D,E*), thus indicating that the EMT involved in this process could be classified into the category of incomplete or transitional EMTs (further referred as tb-EMT), more akin to those described in metastatic processes than to those driving embryonic development (*Kalluri and Weinberg, 2009*; *Lamouille et al., 2014*; *Nieto et al., 2016*; *Pastushenko et al., 2018*). Interestingly, while during gastrulation axial progenitors undergoing EMT through the PS enter differentiation routes (e.g. mesoderm formation), tb-EMT keeps the progenitor properties of these cells, capacitating them to drive further axial extension from the tail bud (*Aires et al., 2019*; *Rodrigo Albors et al., 2018*; *Tzouanacou et al., 2009*). This further indicates that although gastrulation and tail bud formation both involve EMTs on axial progenitors within the epiblast, they seem to be functionally separate processes, suggesting that they should rely on different regulatory mechanisms.

### *Snai1* is required for axial progenitor mobilization to form the tail bud

*Snail* genes are among the most prominent EMT regulators in physiological and pathological processes (*Barrallo-Gimeno and Nieto, 2005*; *Carver et al., 2001*; *Lomelí et al., 2009*; *Murray and*

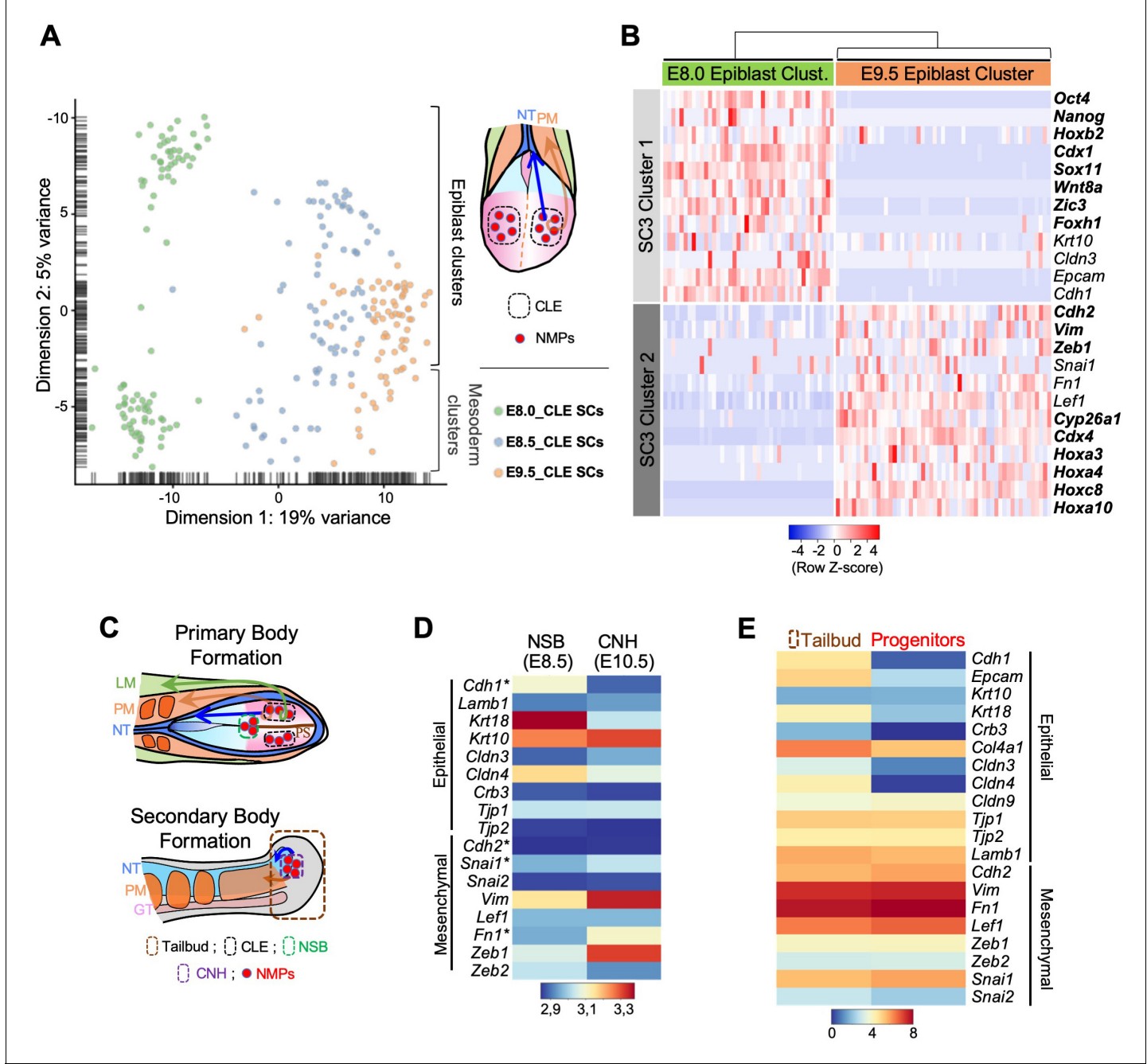

**Figure 1.** Axial progenitors undergo an incomplete EMT during axial extension. (**A**) PCA analysis of CLE scRNA-seq datasets from E8.0, E8.5 and E9.5 embryos. Dimension one represents developmental time, whereas dimension two shows differences in cell characteristics. Epiblast and mesodermal clusters are indicated. During primary body formation, epiblast clusters converge towards the mesodermal compartment. (**B**) Z-score heatmap gene expression analysis of selected genes and marker genes (highlighted in bold), obtained with the SC3 when pre-clustered E8.0 and E9.5 epiblast single-cells are forced to form two clusters. Downregulation of some epithelial-associated genes (e.g. *Cdh1* and *Epcam*) together with upregulation of several mesenchymal-related genes (e.g. *Cdh2*, *Vim*, *Zeb1*) at E9.5 is observed. P-values are shown in *Figure 1—source data 1*. (**C**) Diagrams showing the regions used for the analysis in D and E. (**D**) Comparative heatmap representation of microarray gene expression between E8.5 node-streak border (NSB) and E10.5 chordoneural hinge (CNH). Epithelial and mesenchymal markers are indicated. (**E**) Comparative heatmap representation of RNA-seq gene expression in tail bud axial progenitors and a similar-staged reference total tail bud. Both (**D and E**) analysis indicated that loss of epithelial markers in tail bud axial progenitors is not complete (e.g. *Tjp1* and *Krt10*). NT: Neural Tube; PM: Paraxial Mesoderm; LM: Lateral Mesoderm; GT: Gut; PS: Primitive Streak; * means Average.

The online version of this article includes the following source data and figure supplement(s) for figure 1:

**Source data 1.** p-values corresponding to the SC3 analysis of RNA-seq values represented in *Figure 1* and *Figure 1—figure supplement 1*.

*Figure 1 continued on next page*

*Figure 1 continued*

**Figure supplement 1.** SC3 clustering analysis enables the distinction of epiblast from early mesoderm cells.

*Gridley, 2006*; *Nieto, 2002*; *Zeisberg and Neilson, 2009*). From this gene family, *Snai1* is a good candidate to be involved in EMT processes associated with axial extension, as in mouse embryos it is expressed in the PS region starting at early developmental stages (*Hernández-Martínez et al., 2019*; *Nieto et al., 1992*; *Figure 2—figure supplement 1*). Indeed, gene inactivation experiments in mouse indicated that *Snai1* is involved in EMT processes during gastrulation (*Carver et al., 2001*).

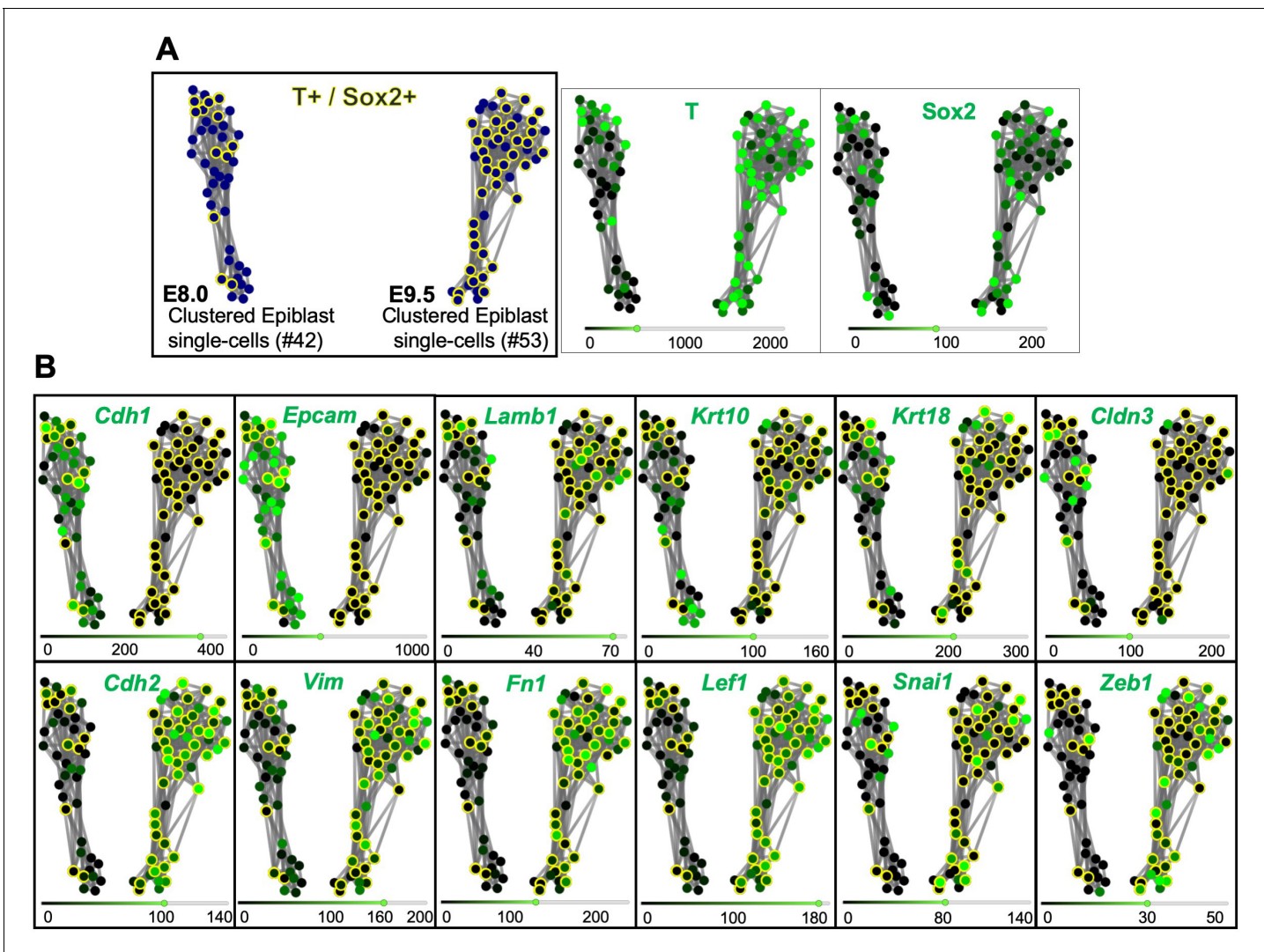

**Figure 2.** *T* and *Sox2* double-positive cells acquire mesenchymal properties during primary body formation. (**A**) RPKM (Reads per kilo base per million mapped reads) SPRING analysis of pre-clustered epiblast E8.0 and E9.5 single cells, organizes them in two clusters according to their developmental stage. *T* and *Sox2* double-positive cells are shown circled in yellow. (**B**) Comparative gene expression analysis of epithelial (upper row) and mesenchymal (lower row) markers focused on *T*+ and *Sox2*+ epiblast cells suggest that they undergo an EMT between E8.0 and E9.5. The scRNA-seq values in RPKM used for this analysis are shown in *Figure 2—source data 1*.

The online version of this article includes the following source data and figure supplement(s) for figure 2:

**Source data 1.** RPKM (Reads per kilo base per million mapped reads) values represented in *Figure 2*, *Figure 2—figure supplement 1* and *Figure 2—figure supplement 2*.

**Figure supplement 1.** *Snai1* is expressed at low levels in regions known to contain axial progenitors.

**Figure supplement 2.** *Tgfbr1* is expressed in regions known to contain axial progenitors.

Interestingly, when this early *Snai1* function was bypassed through a conditional approach that inactivated this gene in the epiblast using the *Meox2-Cre* driver (*Murray and Gridley, 2006*; *Tallquist and Soriano, 2000*) (hereafter termed *Snai1-cKO*), the embryos still failed to develop beyond ~E9.5 (a few embryos survive until E10.5) (*Lomelí et al., 2009*; *Murray and Gridley, 2006*), indicating an essential function of this gene after gastrulation.

Analysis of E9.5 *Snai1-cKO* embryos revealed a fairly well conserved development of trunk structures. In particular, these embryos had a well-defined neural tube, midgut, notochord and a considerable number of somites that, although smaller than those of wild type littermates, retained signs of anterior/posterior compartmentalization (*Figure 3* and *Figure 4Ae-j*). The trunk identity of this part of the *Snai1-cKO* embryos was confirmed by the presence of characteristic intermediate and lateral mesoderm tissues, including forelimb buds, molecular signatures of urogenital system development (*Figure 4Ak,l*) and, in some rare embryos that developed until E10.5, hindlimb buds (*Figure 4Ao,p*). These observations indicate that formation of the primary body is fairly well conserved in *Snai1-cKO* embryos despite *Snai1* being completely absent already at E8.0 (*Murray and Gridley, 2006*) when this region of the body is being laid down.

In contrast, the tail bud of E9.5 *Snai1-cKO* embryos was replaced by a protuberance protruding caudally from the trunk region of the embryo (*Figure 3A,B*, *Figure 3—video 1*, *Figure 4Aa-j* and *Figure 4—video 1*). The finding that in *Snai1-cKO* embryos *Hoxc10* expression was restricted to the protuberance is consistent with this structure replacing the tail bud (*Figure 4Ac-d*). A structure protruding posteriorly from the PS of *Snai1-cKO* embryos was already visible at E8.25 as a bulge (*Figure 5* and *Figure 5—video 1*), when the embryo was building trunk structures. At this stage, *Lfng* expression in *Snai1-cKO* embryos showed variable expression patterns in the region anterior to the bulge, consistent with cycling activity in the presomitic mesoderm generating trunk somites (*Figure 5Ac* and *Figure 5—figure supplement 1*). Interestingly, variable *Lfng* signal was also observed in the bulge. To better characterize the apparently variable *Lfng* expression in the bulge we introduced the LuVeLu reporter transgene (*Aulehla et al., 2008*), which allows live imaging of *Lfng* cycling activity, into the *Snai1-cKO* background. Two-photon live imaging analysis of LuVeLu*:: Snai1-cKO* embryos confirmed the rather normal cyclic activity associated with trunk somite formation and revealed the existence of additional LuVeLu waves in the bulge as a signal moving posteriorly through its dorsal surface (*Figure 6* and *Figure 6—videos 1*, *2*). Therefore, *Lfng* also shows cycling activity in the bulge but with spatial and directional features different from those observed in wild type and more anterior embryonic regions of the mutant embryos, where it runs from posterior to anterior through the more ventrally located presomitic mesoderm. Together, these results indicate that the PS and the bulge represent different functional modules and that *Snai1* is required for secondary but not postcranial primary body formation.

The protuberance observed in *Snai1-cKO* embryos consisted of an epithelial-like layer extending posteriorly from the trunk neural tube, covering a mass of mesenchymal tissue contiguous with the trunk paraxial mesoderm (*Figure 3*, *Figure 3—video 1* and *Figure 5*, *Figure 5—video 1*). At E9.5, the notochord was either bifurcated or had reversed its direction of growth (2 and 6 embryos respectively) becoming associated with gut endodermal tissue that fails to extend into the protuberance and is often detached from the rest of the embryonic tissues (found in 12 from the 19 embryos in which this feature was explored) (*Figure 3Ao-r', Au-v' and B*). At this stage, *Sox2* mRNA expression was detected in the trunk neural tube of *Snai1-cKO* embryos but failed to extend into the epithelial component of the bulge (*Figure 3As-t*). This suggests that, despite histological continuity, the bulge's epithelial sheet is most likely not an extension of the trunk neural tube. Interestingly, at E9.5 this epithelium contained Sox2 protein [likely resulting from significantly higher protein than transcript stability (*Aires et al., 2016*)] in cells also expressing T (*Figure 7*), thus fitting the $T^+/Sox2^+$ NMP molecular signature. The axial progenitor identity of the cells within this epithelium was further supported by the prominent expression of a variety of NMP-related genes (*Aires et al., 2018*; *Rodrigo Albors et al., 2018*; *Gouti et al., 2017*; *Wymeersch et al., 2019*), including T, *Wnt3a*, *Fgf8*, *Nkx1-2* and *Cyp26a*1 (*Figure 3A*). Interestingly, *Fgf8* and *Nkx1-2* expression, as well as $T^+/Sox2^+$ cells, were highly restricted to the bulge's epithelial layer, which contrasts with their extension into the nascent mesenchyme in wild type embryos when they enter the tail bud stage (*Figure 3A* and *Figure 7*). These observations suggest that in the absence of *Snai1*, NMPs become trapped in the epithelial layer of the protuberance, failing to complete the trunk-to-tail transition and disrupting secondary body formation.

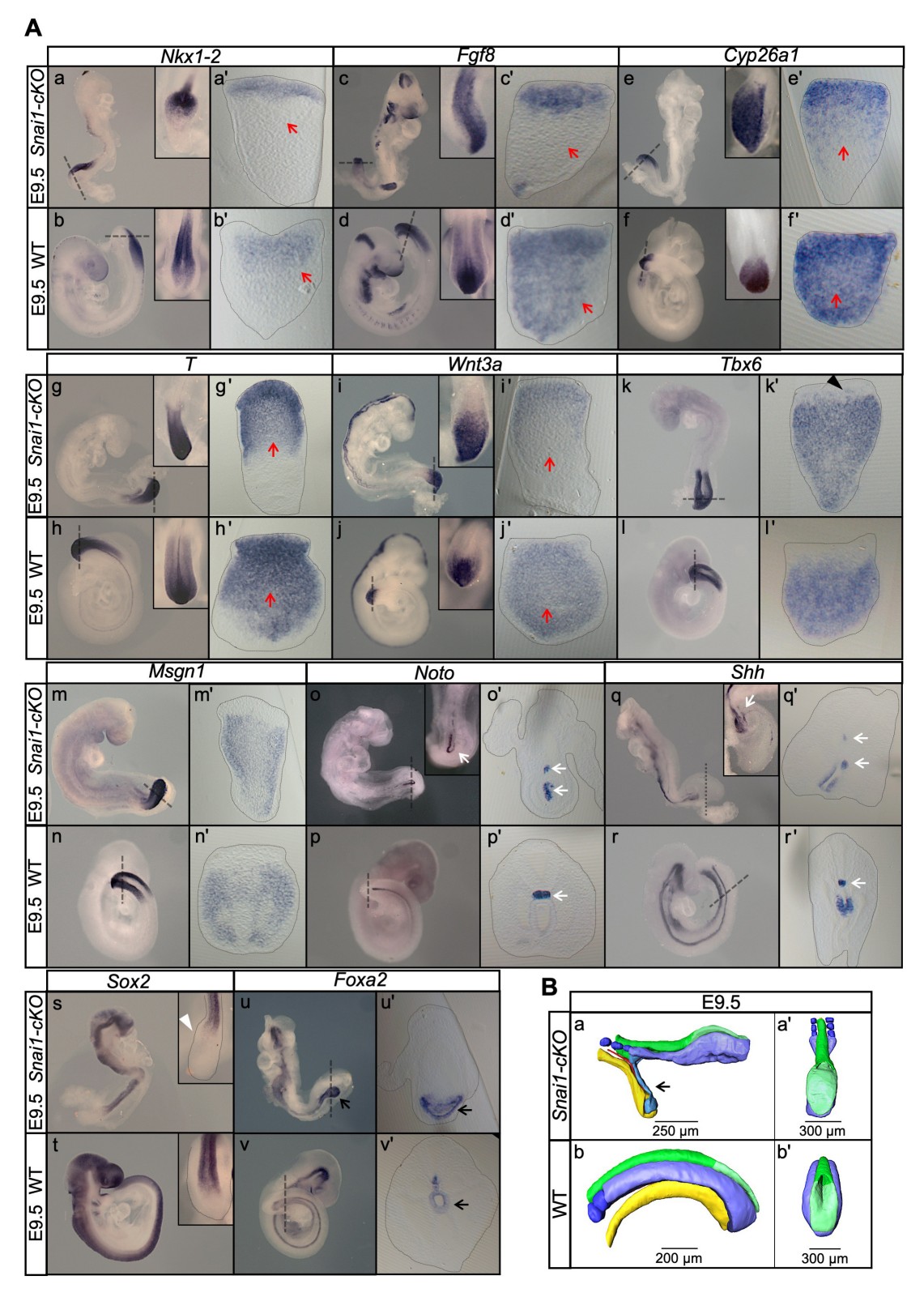

**Figure 3.** Axial extension is disrupted, at the level of the trunk-to-tail transition, in the absence of *Snai1*. (A) Wholemount in situ hybridization with the indicated probes in E9.5 wild type (WT) and *Snai1-cKO embryos.* Axial progenitor-related markers *Fgf8, Nkx1-2, Cyp26a1* were abnormally restricted to the epithelium of the caudal protuberance of *Snai1-cKO* embryos (red arrows) and *T* and *Wnt3a*-stained embryos were substantially downregulated in the central component of the bulge mesenchyme. *Tbx6* expression was observed in the epithelial-like component of the bulge (black arrowhead) in *Figure 3 continued on next page*

Figure 3 continued

addition to its mesenchymal expression. *Msgn1* was present in the presomitic mesoderm and in the mesenchymal component of the bulge. The white arrows in the *Noto*-stained embryos indicate the bifurcated or inverted notochord growth in the absence of *Snai1*. *Sox2* was absent from the bulge region (white arrowhead) of *Snai1-cKO* embryos. Black arrows in the *Foxa2* labelled embryos highlight the abnormal localization of hindgut endoderm in *Snai1-cKO* embryos. (B) 3D reconstructions of E9.5 *Snai1-cKO* and WT posterior/caudal structures: neural tube (green), open epiblast (light green), presomitic mesoderm (light blue), somites (dark blue), notochord (red) and endoderm (yellow). At this stage, the ectopic bulge of *Snai1-cKO* embryos forms a structure that resembles an abnormally extended open epiblast in which the closing neural plate fails to extend caudally. The notochord often bifurcates, with one end following the posterior gut endoderm that is detached from the rest of the embryonic structures (black arrow).

The online version of this article includes the following video for figure 3:

**Figure 3—video 1.** Animated 3D reconstruction of the tail bud of E9.5 embryos.

https://elifesciences.org/articles/56615#fig3video1

The mesenchymal component of the bulge was positive for paraxial mesoderm markers, including *Tbx6* or *Msgn1*, but negative for lateral mesoderm markers such as *Tbx4* (*Figure 3Ak-n'*, *Figure 4A* and *Figure 5Ac-d'*), thus resembling paraxial mesodermal features. Moreover, Tbx6 expression was not restricted to the bulge's mesenchyme but was also observed in the epithelium (*Figure 3Ak-l'*, and *Figure 8B*). Since a portion of tail bud NMPs are positive for *Tbx6* expression (*Javali et al., 2017*), it is possible that the bulge's epithelial component contains progenitor cells that have already acquired some of the mesenchymal traits associated with tail bud axial progenitors. Consistent with this hypothesis, we only detected residual Cdh1 and Epcam expression in the bulge epithelium at E9.5, whereas expression of mesenchymal markers such as Cdh2 and Vim were readily detectable at levels comparable to those in wild type embryos (*Figure 8A* and *Figure 8—figure supplement 1*). In addition, Laminin1 expression in the bulge's epithelium was highly disorganized already at E8.5, contrasting with the characteristic epithelial pattern observed in adjacent more anterior areas of the same embryos or in the epiblast of wild type littermates (*Figure 8B*). Together, these observations indicate that in the absence of *Snai1* axial progenitors initiate tb-EMT and acquire some mesenchymal features but are unable to complete the transition to tail bud development.

## *Snai1* and *Tgfbr1* cooperatively orchestrate the transition from primary to secondary body formation

*Gdf11* has been associated with the trunk-to-tail transition (*Aires et al., 2016*; *Jurberg et al., 2013*; *Liu, 2006*; *Matsubara et al., 2017*). However, its partial functional redundancy with *Gdf8* in this process (*McPherron et al., 2009*; *McPherron et al., 1999*) hinders proper evaluation of its contribution to tb-EMT. Nevertheless, genetic experiments indicate that *Gdf11* (and most likely *Gdf8*) activity in the caudal embryo is mediated by *Tgfbr1* (also known as *Alk5*) (*Andersson et al., 2006*; *Jurberg et al., 2013*), a known EMT regulator (*Derynck et al., 2014*; *Xu et al., 2009*) that is expressed in areas containing axial progenitors (*Figure 2—figure supplement 2*), thus making Gdf11/Tgfbr1 signalling a prime candidate to play a role in tb-EMT. To overcome *Gdf11/Gdf8* functional redundancy, we therefore generated *Tgfbr1* mutant embryos. Initial analysis of *Tgfbr1*[-/-] embryos confirmed the role of this receptor as an inducer of the trunk-to-tail transition (*Figure 9*), and also indicating the absence of redundancy from other type I receptor of the *Tgfβ/BMP* gene family in this process. Indeed, these mutant embryos extended their axis through the trunk region but failed to undergo the transition to tail development, as illustrated by absent hind limb bud markers, lack of *Lin28a* expression at E10.5, which is involved in promoting tail bud axial progenitor activity (*Aires et al., 2019*; *Robinton et al., 2019*), and only residual activation of posterior genes like *Hoxc10* (*Figure 9*). In addition, the anterior-ventral relocation of the allantois -as the embryo turns and starts developing umbilical structures at tail bud stages- was compromised in these mutant embryos and all tested NMP markers (e.g. *T*, *Fgf8* and *Cyp26a1*) seemed to stay associated with dorsal tissues, with no sign of contribution to tail bud mesenchyme/mesodermal tissue (*Figure 9G–L*). Together, these observations are consistent with the embryos finishing their axial extension when *Pou5f1* stops influencing axial progenitor activity derived from its natural decay (*Osorno et al., 2012*), as they are unable to activate the trunk-to-tail transition.

In *Tgfbr1*[-/-] embryos the transition from Cdh1 to Cdh2 in the posterior embryo was very inefficient, as Cdh1 was still detected in the *Tgfbr1*[-/-] epiblast at E9.5 and Cdh2 was only detected at very low levels (*Figure 10A*). Concomitantly, we observed the characteristic Laminin1 epithelial patterns,

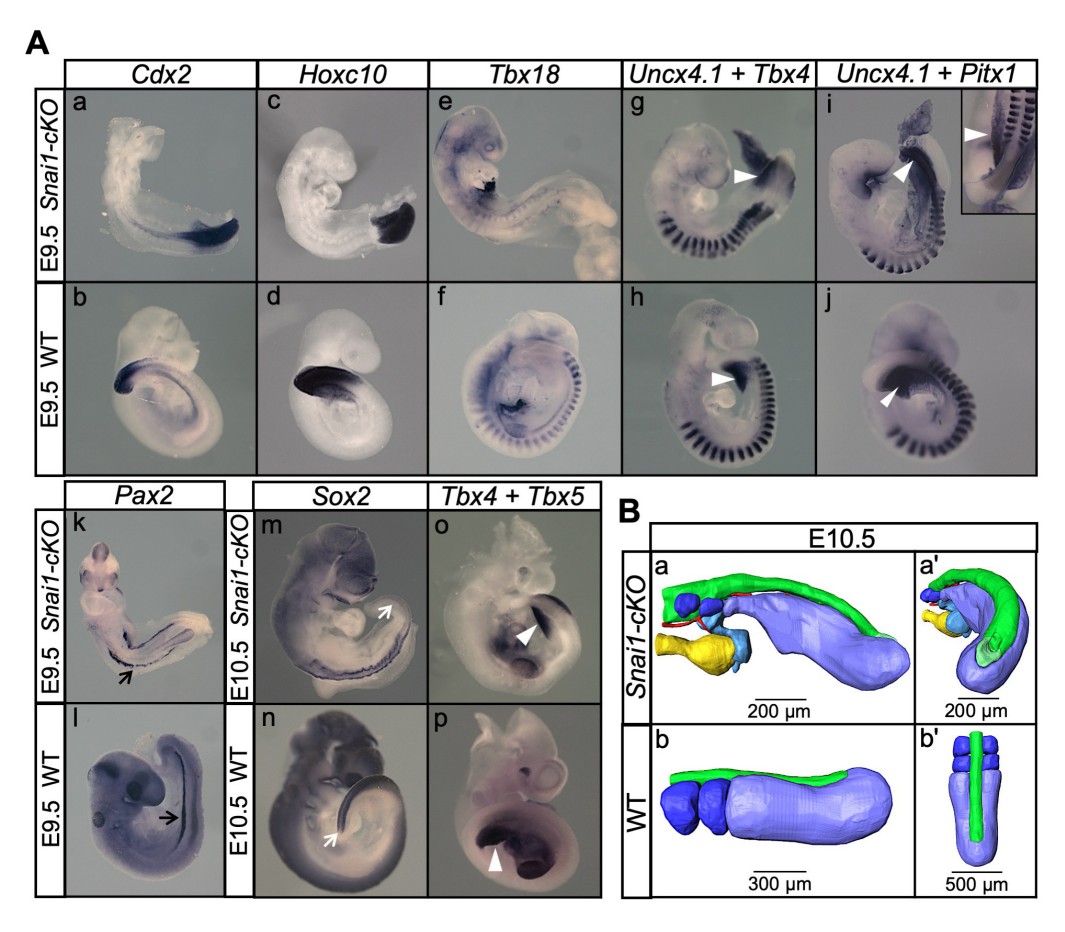

**Figure 4.** *Snai1-cKO* embryos develop fairly-well organized primary body structures but are unable to form a tail bud. (**A**) Wholemount in situ hybridization with the indicated markers in wild type (WT) and *Snai1-cKO* mutant embryos at E9.5 and E10.5. The insets show close ups of the caudal region. Transverse sections at the levels indicated in the whole-mount embryo are also shown. *Cdx2* and *Hoxc10* expression in *Snai1-cKO* embryos indicate that the bulge is the equivalent of the tail bud in similar stage wild type embryos. The black arrows in the *Pax2*-stained embryos indicate the presence of intermediate mesoderm in both *Snai1-cKO* and wild type embryos. At E9.5, *Tbx18* expression and the combined in situ hybridization for *Uncx4.1/Tbx4* and *Uncx4.1/Pitx1*, indicate the presence of fairly-well organized trunk somites and hindlimb buds (white arrowheads) in *Snai1-cKO* embryos, which is also highlighted in some mutant embryos, that survived at E10.5, with *Tbx5* and *Tbx4* expression. White arrows indicate that *Sox2* is still downregulated in the bulge of E10.5 *Snai1-cKO* embryos, in comparison to the tail bud of wild type littermates. (**B**) 3D reconstructions of caudal structures of E10.5 wild type and *Snai1-cKO* embryos. Wild type embryos have a closed neural tube (in green), dorsal to the presomitic mesoderm (in blue) and somites (in dark blue). In contrast, *Snai1-cKO* embryos have a bifurcated notochord (red), detached endoderm (yellow) and the bulge resembles a still open caudal epiblast (in green). Somites are shown in dark blue and extra mesoderm in *Snai1-cKO* embryos is highlighted in cyan. Note that the structures posterior to the last somites in *Snai1-cKO* embryos did not extend further than at E9.5, leading to a shorter axis.

The online version of this article includes the following video for figure 4:

**Figure 4—video 1.** Animated 3D reconstruction of the tail bud of E10.5 embryos.

https://elifesciences.org/articles/56615#fig4video1

as well as higher Epcam and lower Vim levels in the *Tgfbr1* mutant epiblast than in similar stage control embryos (***Figure 10B*** and ***Figure 10—figure supplement 1***). These data support an essential role for Gdf11/Tgfbr1 signalling in triggering tb-EMT and suggest that this signalling activity precedes that of *Snai1* during tb-EMT. However, contrary to what has been described in other experimental contexts (***Lamouille et al., 2014***), *Snai1* is most likely not a direct target of Tgfbr1 signalling during tb-EMT, as we did not find any evidence for increased *Snai1* expression in the progenitor-containing region of *Tgfbr1*-overexpressing embryos [*Cdx2-Alk5*$^{CA}$ transgenics (***Jurberg et al., 2013***; ***Figure 11B***)]. This suggests that *Snai1* and Gdf11/Tgfbr1 signalling are part of independent pathways that converge to orchestrate the tb-EMT. Indeed, *Snai1* expression in the PS region covers

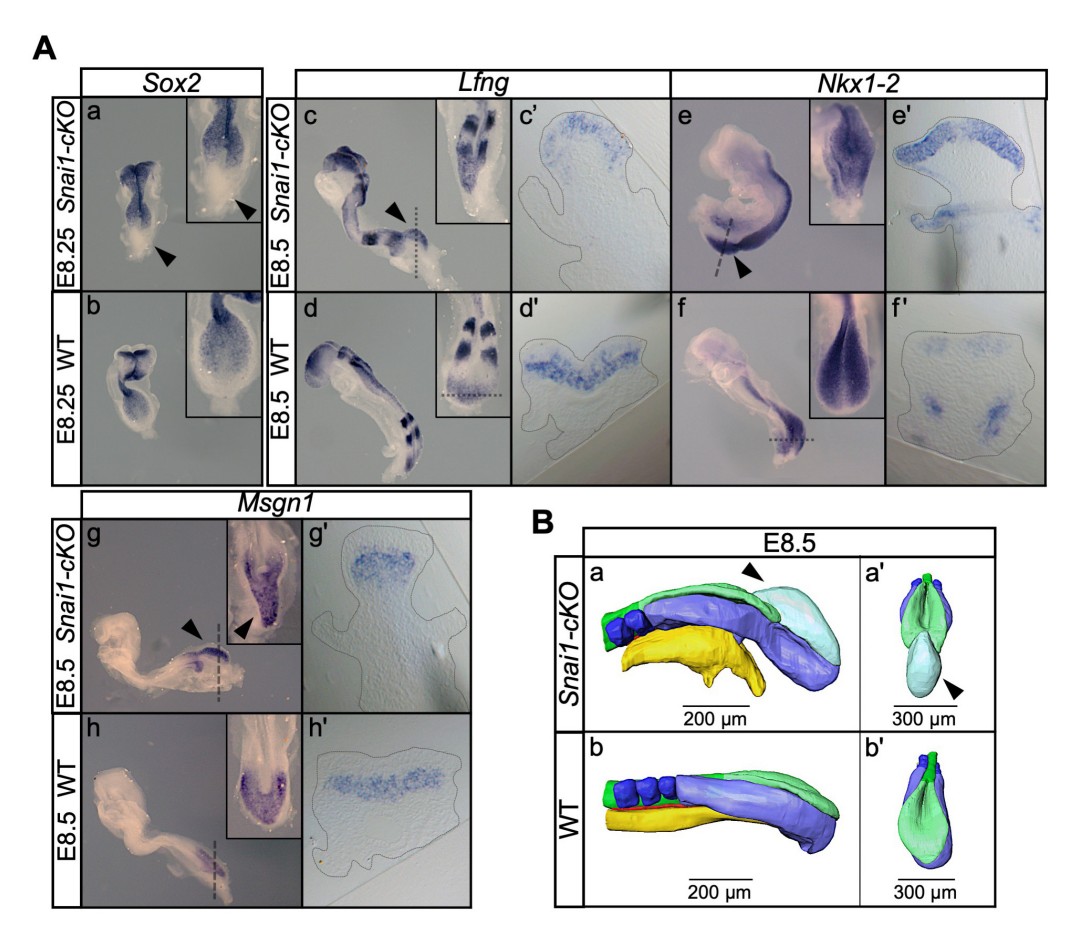

**Figure 5.** *Snai1-cKO* embryos develop an ectopic bulge associated with the PS. (**A**) Wholemount in situ hybridization with the indicated markers in wild type (WT) and *Snai1-cKO* mutant embryos at E8.5. The inlets show close ups of the caudal region. Transverse sections at the levels indicated in the whole-mount embryo are also shown. In the absence of *Snai1*, a protuberance (black arrowheads) starts to arise from the PS around E8.25. This ectopic bulge is positive for *Lfng* and *Nkx1-2*, but not for *Sox2*. *Lfng* and *Nkx1-2* expression are restricted to the epithelial-part of the bulge. *Msng1* was only found in the mesenchymal component. (**B**) 3D reconstruction of E8.5 *Snai1-cKO* and wild type posterior/caudal structures: neural tube (green), open epiblast (light green), presomitic mesoderm (light blue), somites (dark blue), notochord (red) and endoderm (yellow). At E8.5, *Snai1-cKO* embryos contain an ectopic bulge (black arrowhead) associated with the PS and the hindgut endoderm shows abnormal development.

The online version of this article includes the following video and figure supplement(s) for figure 5:

**Figure supplement 1.** *Lfng* dynamic activity in the bulge of *Snai1-cKO* embryos.

**Figure 5—video 1.** Animated 3D reconstruction of the tail bud of E8.5 embryos.

https://elifesciences.org/articles/56615#fig5video1

only a small region within the *Gdf11* expression domain in the epiblast (*Hernández-Martínez et al., 2019*; *Nieto et al., 1992*; *Figure 11A*). Therefore, it is possible that *Snai1* marks a subset of axial progenitors within the cell pool exposed to Gdf11/Tgfbr1 signalling to be recruited to the tail bud by entering tb-EMT.

## Extended *Snai1* expression in the PS mobilizes functional tail bud progenitors

A prediction from this hypothesis is that expanding the *Snai1* expression domain in the PS would lead to an increase in the number of cells entering the tail bud route. We tested this hypothesis using a transgenic approach expressing *Snai1* under the control of the *T-str* enhancer (*Clements et al., 1996*), a regulatory element of the *Brachyury* gene that drives expression in progenitor cells within the PS (*Anderson et al., 2013*). About half of these transgenic embryos (n = 43)

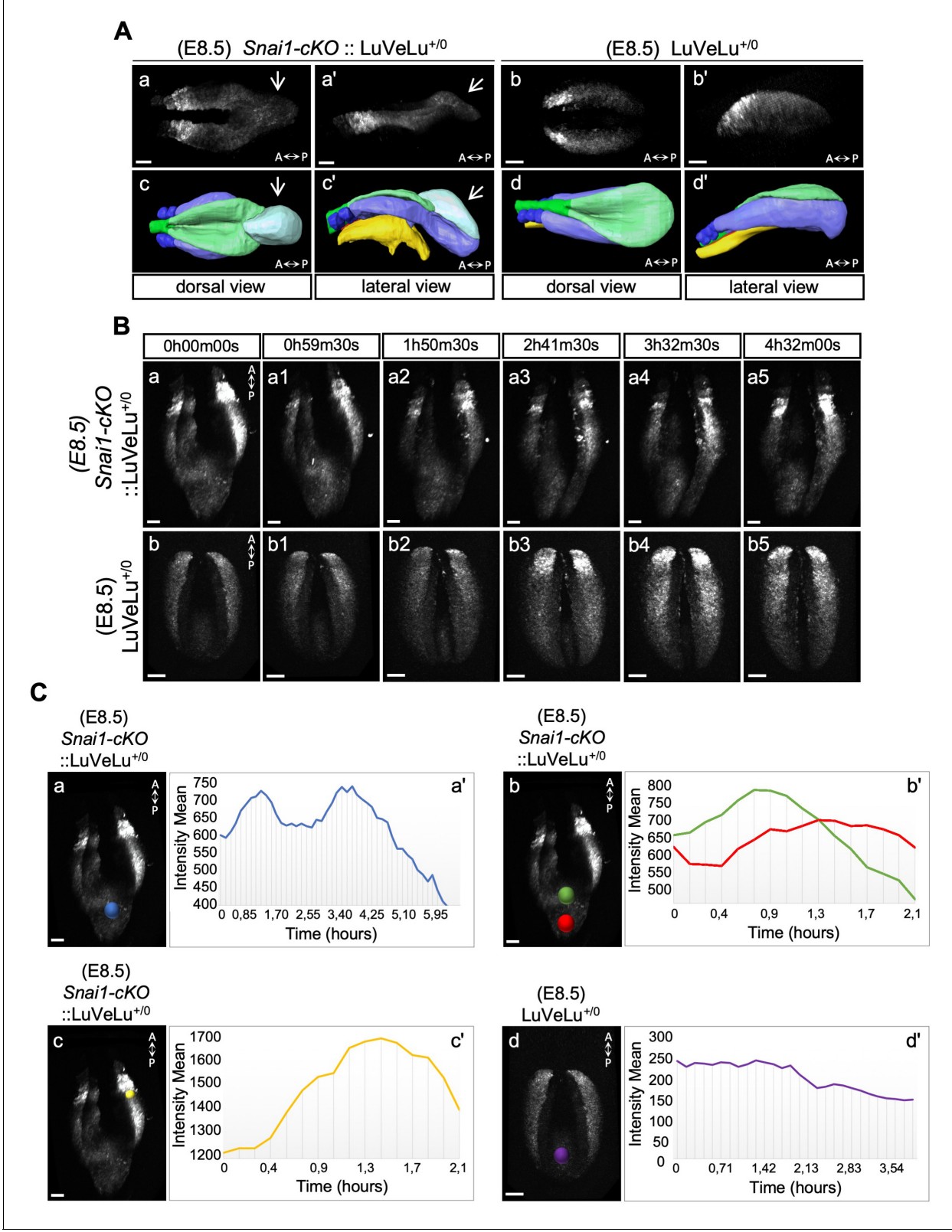

**Figure 6.** 3D two-photon live imaging of LuVeLu reporter expression in E8.5 *Snai1-cKO* and control embryos. (**A**) Snapshot at time-point = 0 of the LuVeLu reporter, in *Snai1-cKO* (from **Figure 6—video 1**) and control (from **Figure 6—video 2**) E8.5 embryos. In addition to the normal LuVeLu signal in the presomitic mesoderm, *Snai1-cKO* embryos display ectopic LuVeLu expression in the epithelial-like component of the bulge. (**B**) Snapshots from **Figure 6—video 1** and **Figure 6—video 2**, of time-lapsed two-photon live imaging of the LuVeLu reporter in E8.5 *Snai1-cKO* and control embryos at

*Figure 6 continued on next page*

*Figure 6 continued*

the indicated time points. *Snai1-cKO* embryos have ectopic LuVeLu expression in the epithelial-like component of the bulge in addition to the normal LuVeLu signal in the presomitic mesoderm. (C) Quantitative analysis of LuVeLu cycling activity in the bulge of *Snai1-cKO* embryos. Intensity mean was calculated in the region highlighted by the blue spot (Ca) and plotted for each imaged time-point (8.5 min interval; Ca'). The existence of two-peaks (at t = 1.3 hr and t = 3.6 hr of the time-lapse) and a substantial decrease between them, suggests cycling activity in the bulge of the mutant embryo. These waves occur from anterior to posterior, once the higher intensity mean peak observed in the posterior part of the bulge (red spot; Cb) occurred later than the higher peak measured in the anterior part of the bulge (green spot; Cb). The higher measured wave peak in the posterior part of the bulge (red spot; Cb) coincides with the time-point corresponding to the higher intensity mean peak detected in the newly formed somite (yellow spot; Cc). No signs of LuVeLu cycling activity were observed near the PS (purple spot; Cd) in LuVeLu$^{+/0}$ control embryos. Scale bar: 50 μm.

The online version of this article includes the following video(s) for figure 6:

**Figure 6—video 1.** LuVeLu activity in a E8.5 *Snai1-cKO* embryo.
https://elifesciences.org/articles/56615#fig6video1
**Figure 6—video 2.** LuVeLu activity in a E8.5 wild type embryo.
https://elifesciences.org/articles/56615#fig6video2

showed caudal morphological abnormalities at E9.5, including axial shortening and premature closure of the caudal neuropore (*Figure 12*). In these embryos the *T*, *Cyp26a* and *Nkx1.2* domains expanded into the ventral mesenchymal region of the shortened tail bud (*Figure 12C-D2, I-L2*), consistent with production and delamination of tail bud axial progenitors in regions more anterior to the normal position of the tail bud. In addition, these embryos had a variable number of ectopic neural tubes connected caudally with the expanded *Nkx1.2*-positive mesenchymal domain, aligning ventrally to the main spinal cord (*Figure 12*), which is consistent with these cells having acquired tail bud NMP identity. We also found cells expressing *T* and *Tbx6* intermingled with these ectopic neural tubes (*Figure 12C1–C2, M1*), further supporting the NMP identity of these progenitors. Interestingly, these extra neural tubes contained a degree of dorso-ventral patterning, as shown by *Foxa2* expression in their ventral domain, corresponding to the floor plate (*Figure 12O1*), and by *Wnt3a* expression in the dorsal part of some of these ectopic neural tubes (*Figure 12G1*). Together, these results indicate that *Snai1* is necessary and sufficient for axial progenitor mobilization during tail bud formation.

## Discussion

The transition from primary to secondary body formation involves at least two coordinated but independent processes. One such process entails activation of a terminal differentiation program in the progenitors for the intermediate and lateral components of the mesoderm that results in the induction of the hindlimb bud and of the mesodermal component of the cloaca (*Jurberg et al., 2013*). The second process involves the common progenitors for the spinal cord and paraxial mesoderm that become the tail bud, from where they promote further extension of the body axis to generate post-sacral body structures (*Henrique et al., 2015*; *Steventon and Martinez Arias, 2017*; *Wilson et al., 2009*).

### *Snai1* and *Tgfbr1* play a key role in the gene regulatory networks driving tail bud formation

Genetic experiments have shown that *Gdf11* plays an important role in both processes by activating signalling in the axial progenitors through the Tgfbr1 receptor. Indeed, premature activation of this signalling in axial progenitors anticipates the trunk-to-tail transition (*Jurberg et al., 2013*; *Liu, 2006*; *Matsubara et al., 2017*), and *Tgfbr1* null embryos are unable to engage in tail development (this work).

A variety of experimental evidence indicates that *Isl1* is a key component of the network downstream of *Tgfbr1* to induce hindlimb and cloacal tissues (*Jurberg et al., 2013*). The same experiments also showed that this gene is not involved in tail bud formation, indicating the existence of a different mechanism regulating the formation of this structure. Here we presented evidence for the critical role of *Snai1* in this process. Interestingly, however, this gene is not under the control of Tgfbr1 signalling but provides instead a parallel activity that promotes segregation of axial progenitors into the future tail bud upon functional convergence with Tgfbr1 signalling. The *Snai1-cKO*

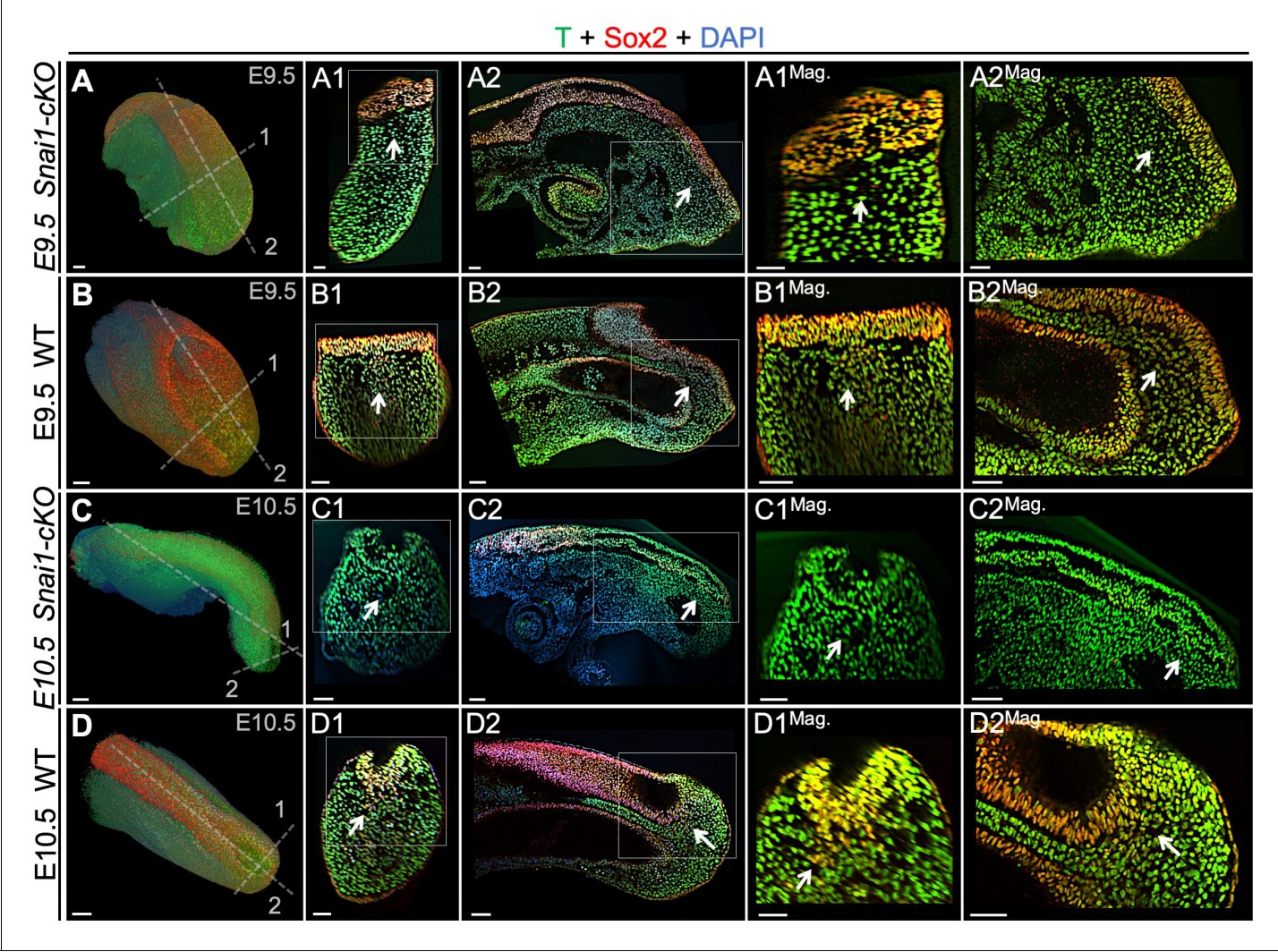

**Figure 7.** T and Sox2 double-positive cells are restricted to the epiblast of *Snai1-cKO* embryos. Wholemount immunostaining for T (green), Sox2 (red) and DAPI (blue) in E9.5 (**A** and **B**) series) and E10.5 (**C** and **D**) series) *Snai1-cKO* and wild type (WT) embryos. Transversal and sagittal optical sections together with magnifications (Mag.) are also shown. In E9.5 wild type embryos, T$^+$/Sox2$^+$ cells are found in the mesenchyme below the epiblast (white arrows). Conversely, in similar staged *Snai1-cKO* embryos, they are restricted to the epithelial component of the bulge. At E10.5 most of the bulge's epithelium, of *Snai1-cKO* embryos, closed into a tube that is mostly negative for Sox2 and positive for T. In contrast, T and Sox2 double-positive cells are still found in mesenchymal compartments of the tail bud of wild type embryos. Magnifications are shown without DAPI. Scale bars correspond to 50 μm.

mutant phenotype indicates that tail bud formation starts rather early in development when the embryo is still actively engaged in PS-mediated trunk development, much earlier than the appearance of morphological signs of a tail bud. This phenotype thus uncovers the existence of a functional developmental module specifically devoted to the formation of the tail bud, independent from that involved in trunk formation. The existence of such 'tail bud module' somehow resembles hindlimb or cloacal induction, thus suggesting a unifying mode by which Gdf11/Tgfbr1 signalling activity triggers the different processes involved in the trunk-to-tail transition, consisting in activating alternative developmental programs in different subsets of axial progenitors. The observation that NMPs are suppressed by *Isl1* activity (*Jurberg et al., 2013*) indicates that the intrinsic properties of these cells differ from those of the progenitors for intermediate and lateral mesoderm, determining whether they engage in tail bud or hindlimb/cloaca formation upon *Tgfbr1* functional input.

Different experimental observations indicate that trunk length and the position of the switch from trunk to tail developmental modes result from quantitative functional balance between *Gdf11* and

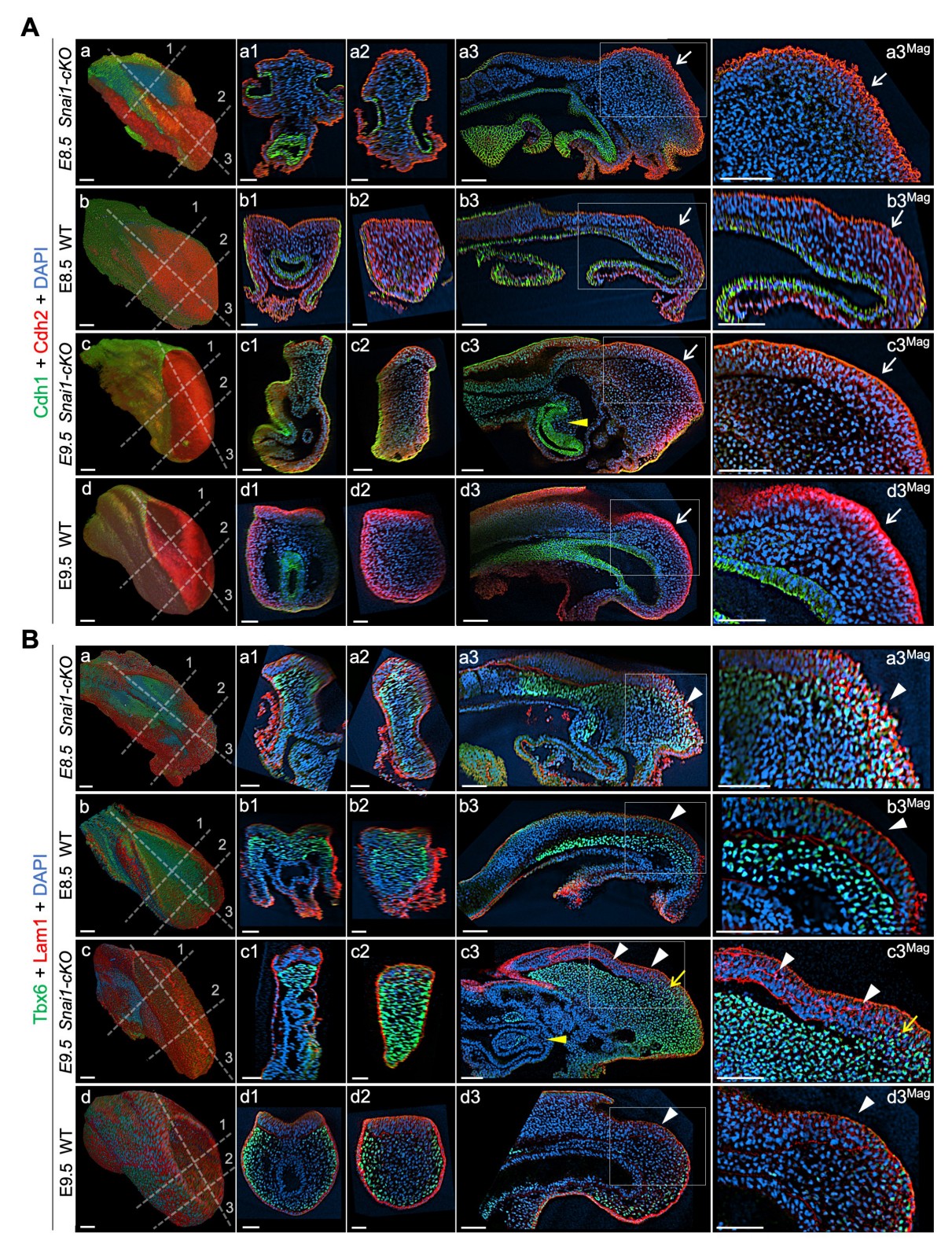

**Figure 8.** Axial progenitors acquire mesenchymal properties in the absence of *Snai1*. Wholemount immunostainings for Cdh1 (green) plus Cdh2 (red) (**A**) and Tbx6 (green) plus Lam1 (red) (**B**), in E8.5 and E9.5 *Snai1-cKO* and wild type (WT) embryos. Transversal and sagittal optical sections in the indicated regions are also shown together with magnifications (Mag.). The transition from Cdh1 to Cdh2 still occurs in the epithelial component of the bulge in the absence of *Snai1* (white arrows). This region of *Snai1-cKO* embryos contains ectopic Tbx6-positive cells (yellow arrow) and a severely

*Figure 8 continued on next page*

*Figure 8 continued*

disorganized Lam1 expression (white arrowheads). Yellow arrowheads highlight posterior gut bifurcation in E9.5 *Snai1-cKO* embryos. Scale bars: 50 µm. DAPI staining in blue.

The online version of this article includes the following figure supplement(s) for figure 8:

**Figure supplement 1.** Wholemount immunohistochemistry in wild type and *Snai1-cKO* embryos.

*Pou5f1* activities (*Aires et al., 2016*; *DeVeale et al., 2013*; *Jurberg et al., 2013*; *Mallo, 2018*). These observations, together with the *Snai1-cKO* mutant phenotype lead us to suggest a model for axial extension whereby at early stages (mouse E8.0) progenitor activity is dominated by *Pou5f1*, keeping an active PS, while *Gdf11/Tgfbr1* functional contribution is negligible or extremely small. As

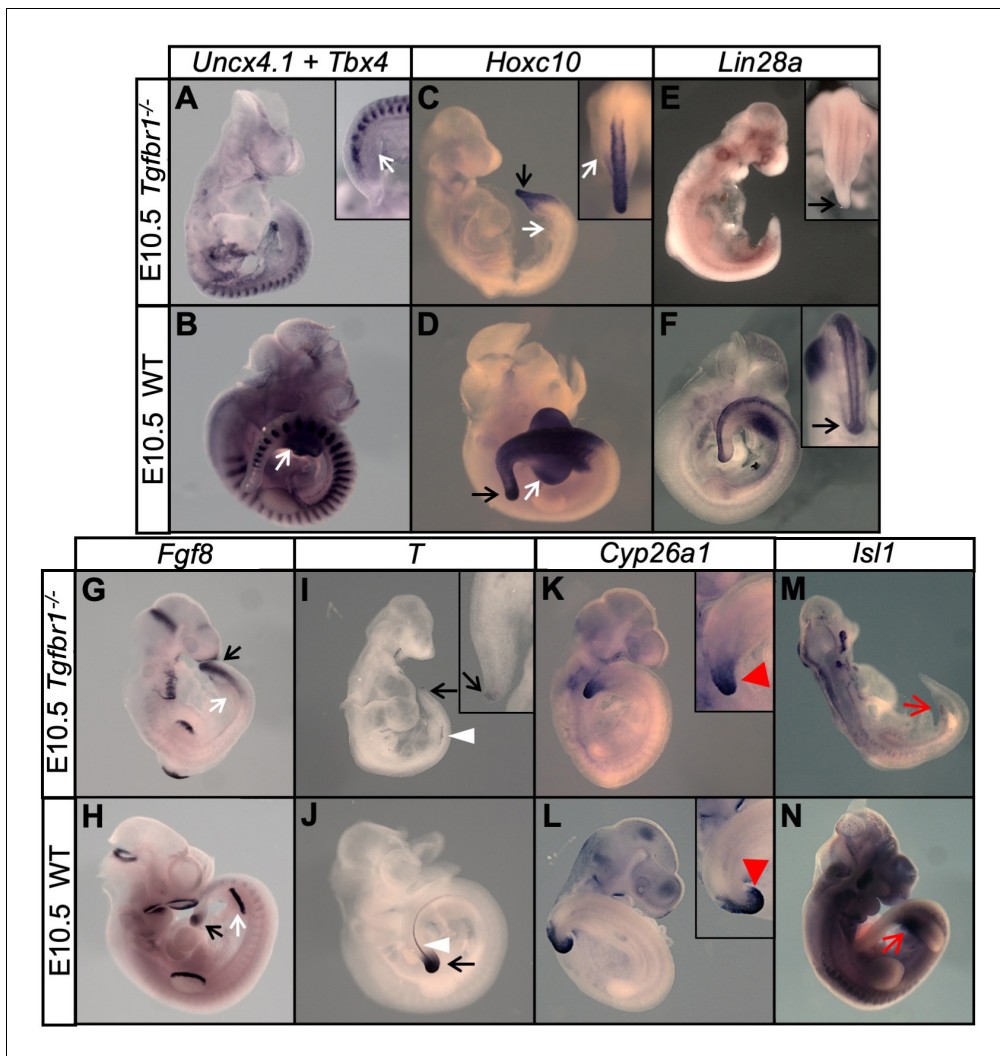

**Figure 9.** *Tgfbr1* mutant embryos are unable to undergo the trunk-to-tail transition. (**A–N**) Wholemount in situ hybridization with the indicated markers in E10.5 *Tgfbr1⁻/⁻* and wild type (WT) embryos. Absent hindlimb bud formation in *Tgfbr1* mutants is highlighted by absent *Tbx4*, *Fgf8* and *Hoxc10* expression (**A–D, G and H**) in the relevant area (white arrows). Tail bud formation is deficient in *Tgfbr1* mutant embryos (black arrows). *Lin28a* and *T* are drastically downregulated in the tail of *Tgfbr1* mutant embryos (black arrows in **E** and **I**), suggesting that these mutants are caudally truncated at this stage. The white arrowheads indicate the notochord, which did not continue its development to more caudal regions in *Tgfbr1* mutant embryos. Red arrowheads highlight the abnormal *Cyp26a1* expression in the *Tgfbr1⁻/⁻* tail. Red arrows in **M**) and **N**) indicate absent *Isl1* expression in the prospective developing cloaca of *Tgfbr1* mutant embryos.

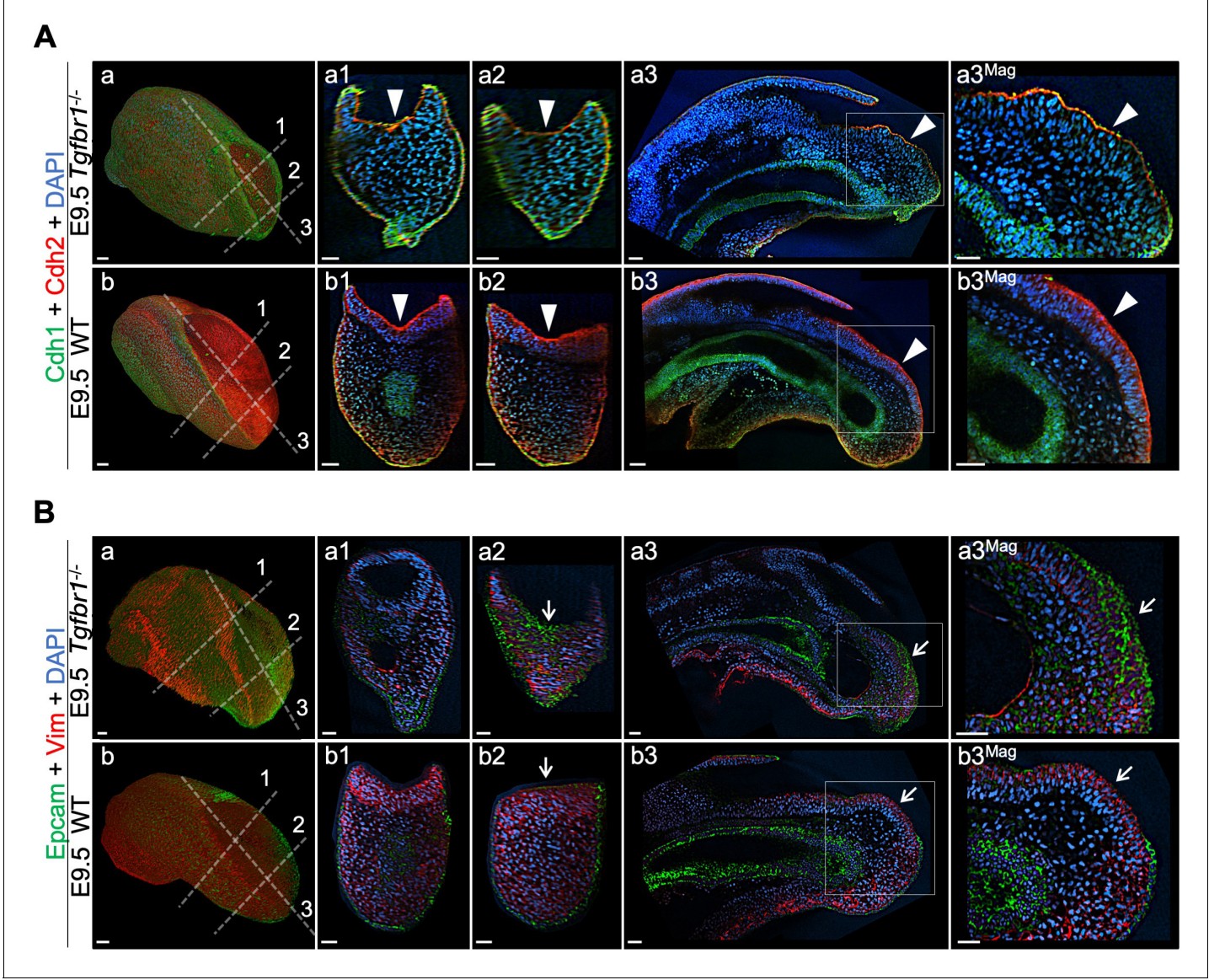

**Figure 10.** Tail bud EMT is compromised in the absence of *Tgfbr1*. Wholemount immunostaining for Cdh1 (green) plus Cdh2 (red) (A) and Epcam (green) plus Vim (red) (B) in *Tgfbr1*⁻/⁻ and wild type (WT) embryos. Transversal and sagittal optical sections through the indicated regions are also shown together with magnifications (Mag.). White arrowheads indicate incomplete Cdh1 to Cdh2 switch in *Tgfbr1* mutants. White arrows show persistent Epcam and deficient Vim expression in the epithelium of the mutant embryos. DAPI staining is shown in blue. Scale bars: 50 μm.

The online version of this article includes the following figure supplement(s) for figure 10:

**Figure supplement 1.** Whole mount immunohistochemistry in wild type and *Tgfbr1* embryos.

development proceeds *Pou5f1* activity is progressively reduced in the epiblast (*Osorno et al., 2012*) and *Tgfbr1* acquires increased functional weight in this area. *Snai1*-positive cells in the PS are the first to respond to the reversed Pou5f1/Tgfbr1 activity balance, resulting in NMPs mobilizing to the tail bud. This process is already triggered when the PS still actively coordinates development at more anterior levels, indicated by the *Snai1-cKO* phenotype. As development proceeds, *Pou5f1* activity decreases (*Osorno et al., 2012*), and given its requirement for PS maintenance (*Aires et al., 2016*; *DeVeale et al., 2013*), it is likely to be responsible, together with *Snai1,* for PS regression and caudal epiblast closure. Under these conditions, *Tgfbr1* eventually takes full control of axial progenitor activity, leaving the tail bud as the only source of axial extension activity and recruiting lateral mesoderm progenitors into hindlimb and cloacal structures. In addition, this model implies an

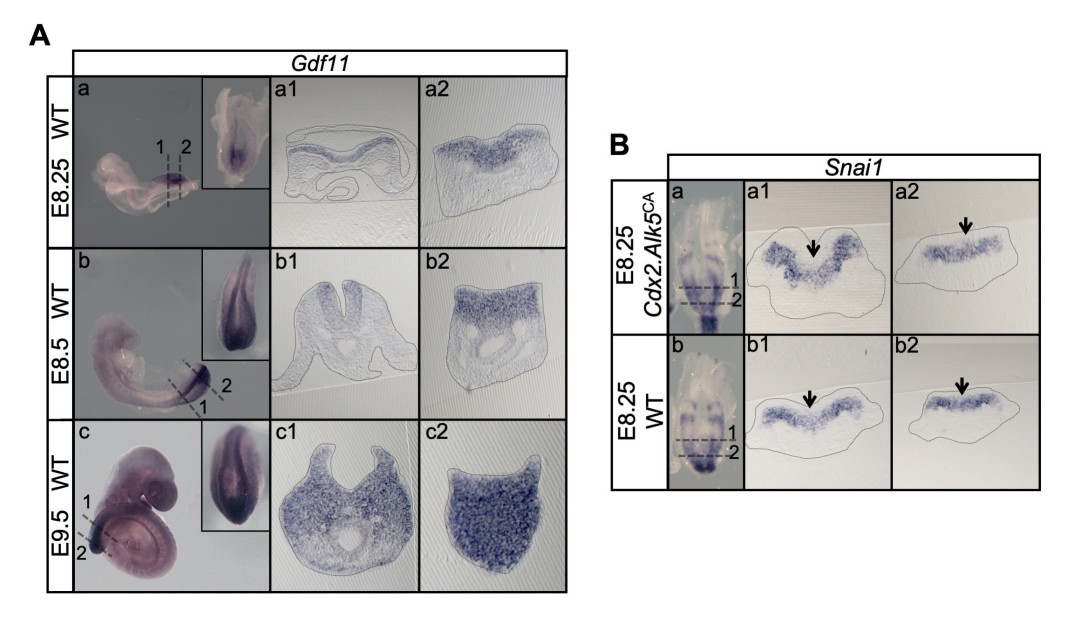

**Figure 11.** *Snai1* is not a downstream target of *Tgfbr1*. (A) Wholemount in situ hybridization for *Gdf11* in E8.25, E8.5 and E9.5 wild type (WT) embryos. Transversal sections through the areas indicated in the whole mounted embryo highlight the increase of *Gdf11* in caudal tissues during primary body formation. (B) Overexpression of a constitutively active form of *Tgfbr1* (transgenic *Cdx2-Alk5^CA*) does not result in increased *Snai1* expression as shown in transversal sections of the areas indicated in the wholemount embryo. Arrows indicate the position of the PS, where low levels of *Snai1* expression can be observed both in the WT (Bb) and transgenic embryos (Ba).

overlap of primary and secondary processes of neural tube formation, and therefore fits better with classical observations in human embryos (*Saitsu et al., 2004*). A failure in the coordination between these two processes can ultimately lead to the generation of spina bifida (*Saitsu and Shiota, 2008*).

## Two different functional types of EMT are required during vertebrate axial extension

Here we have shown that the transition from primary to secondary body formation in the mouse entails an incomplete EMT acting on axial progenitors and termed this new developmental process 'tail bud EMT' (tb-EMT), as it is required for tail bud formation. This EMT is different from that driving epiblast progenitors through the PS to generate mesodermal tissues during gastrulation in different ways. (1) progenitors undergoing tb-EMT do not enter a differentiation route but retain instead progenitor properties that endow them with the capacity to further extend the body axis by generating both the neural tube and paraxial mesoderm of the tail. (2) tb-EMT is incomplete, keeping expression of a subset of epithelial markers that leave the progenitors in a transitional state. This property might actually facilitate the production of both neural and mesodermal structures from tail bud axial progenitors. Moreover, it is possible that maintenance of some epithelial properties can help progenitors to keep contact with the trunk neural epithelium that they will eventually extend by regaining full epithelial features. Conversely, these cells would be able to enter mesodermal differentiation routes just by completing the EMT, thus circumventing the need for a functional PS. Interestingly, comparison of CNH and tail bud mesenchyme heterochronic grafts (*Cambray and Wilson, 2002*) showed that after tail bud progenitors enter the mesodermal compartment, they have no longer the potential to generate neural tissues, becoming restricted to mesodermal compartments, further supporting that tail bud progenitors are kept in a transitory state instead of acquiring full mesenchymal characteristics when entering the tail bud.

Another major difference between gastrulation- and tail bud- associated EMTs resides in their genetic control. Genetic experiments identified a number of genes involved in early gastrulation steps, including *Crumbs2*, *p120-catenin*, *Eomes*, *Nodal* and FGF signalling (*Arnold et al., 2008*; *Deng et al., 1994*; *Hernández-Martínez et al., 2019*; *Ramkumar et al., 2016*; *Voiculescu et al.,*

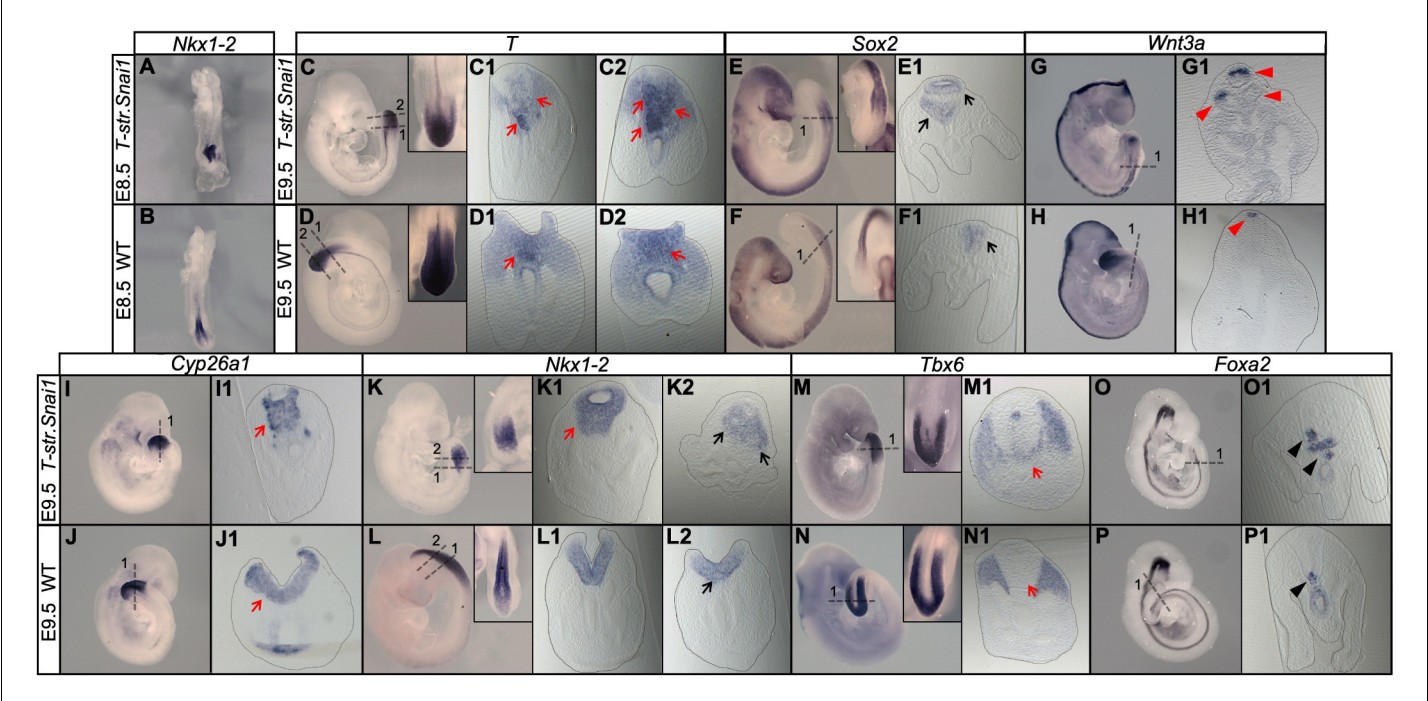

**Figure 12.** Extended *Snai1* expression in the PS is sufficient to mobilize axial progenitors from the epiblast. (**A–R**) Wholemount in situ hybridization for the indicated markers in wild type (WT) and transgenic *T-str-Snai1* embryos. *Nkx1-2* expression in the transgenic embryos (**A**) indicates an already deformed caudal epiblast at E8.5. At E9.5 *T-str-Snai1* embryos are caudally truncated and display a complete premature closure of their caudal epiblast. Red arrows in C2, I1 and K1 highlight the ventrally extended *T*, *Cyp26a1* and *Nkx1-2* expression, indicating ectopic mobilization of axial progenitors. Black arrows in E1 and K2 show ventrally located ectopic neural tubes in *T-str-Snai1* embryos. Red arrows in C1 and M1 indicate *T*, *Tbx6*-positive cells intermingled with the ectopic neural tubes. The black arrowheads in O1 and P1 indicate the floor plate, showing conserved ventral patterning in ectopic neural tubes of *T-str.Snai1* embryos. The red arrowheads in G1 show *Wnt3a* expression in the dorsal region of the ectopic neural tubes.

*2014*; *Yamaguchi et al., 1994*). *Tgfbr1*, which is an essential component of tb-EMT, is clearly not involved in gastrulation as embryos lacking this receptor can go through this developmental step and remarkably generate trunk structures. *Snai1* seems to be required for both gastrulation and tb-EMT, although its role might be different in both processes. *Snai1* null mutant embryos generate mesoderm during gastrulation but the mesodermal cells produced are unable to complete their EMT process, keeping several epithelial characteristics and failing to downregulate Cdh1 (*Carver et al., 2001*). During tb-EMT cells lacking *Snai1* had already lost several epithelial characteristics, most likely resulting from a previous *Tgfbr1* functional input, suggesting that its role during tail bud formation must be different from that in gastrulation. In *Snai1-cKO* embryos, axial progenitors were restricted to epithelial-like layer of the bulge contiguous with the posterior epiblast. Conversely, these cells seem to leave the epiblast prematurely in *T-str-Snai1* transgenic embryos. Therefore, although the role of *Snai1* in tail bud formation requires further evaluation, it is possible that it might be involved in cell mobilization. Interestingly, the phenotypes of both *Snai1* null and of *Snai1-cKO* mutant embryos suggest that PS activity during postcranial elongation is functionally different to that during early gastrulation because *Snai1-cKO* embryos are able to efficiently elongate the postcranial primary body even in the absence of detectable *Snai1* (*Murray and Gridley, 2006*). This observation is in keeping with a large body of genetic data indicating differences in the gene networks regulating early gastrulation processes and postcranial axial elongation (*Andre et al., 2015*; *Herrmann et al., 1990*; *Liu et al., 1999*; *Savory et al., 2011*; *Takada et al., 1994*).

It should be also noted that tb-EMT functional and molecular characteristics are more akin to those described for the transitions involved in metastatic processes than to the typical developmental EMTs (e.g. gastrulation) (*Acloque et al., 2009*; *Kalluri and Weinberg, 2009*; *Lamouille et al., 2014*; *Nieto et al., 2016*). These transient tail bud progenitors might not only be used to re-assess

neural tube closure defects, but also represent a novel in vivo model to study the mechanisms activating *Tgfbr1/Snai1*-dependent metastatic processes.

# Materials and methods

## Key resources table

| Reagent type (species) or resource | Designation | Source or reference | Identifiers | Additional information |
|---|---|---|---|---|
| Gene (*M. musculus*) | *Snai1* | MGI | MGI:98330 | (other names) *Snail* |
| Gene (*M. musculus*) | *Tgfbr1* | MGI | MGI:98728 | (other names) *Alk5* |
| Genetic reagent (*M. musculus*) | B6.1239S4-Meox2tm1 (cre)Sor/J | Jackson Labs | Stock No 003755 RRID:IMSR_JAX:026858 | *Tallquist and Soriano, 2000* (another name Meox2-Cre$^{+/0}$) |
| Genetic reagent (*M. musculus*) | B6.129S-Snai1tm2Grid/J | Jackson Labs | Stock No 010686 RRID:IMSR_JAX:010686 | *Murray and Gridley, 2006* (another name Snai1$^{flox/flox}$) |
| Genetic reagent (*M. musculus*) | LuVeLu | *Aulehla et al., 2008* | | Transgenic |
| Genetic reagent (*M. musculus*) | *T-str-Snai1* | This paper | | Transgenic |
| Genetic reagent (*M. musculus*) | *Tgfbr1$^{+/-}$* | This paper | | Targeted null mutation |
| Genetic reagent (*M. musculus*) | *Cdx2-Alk5$^{CA}$* | *Jurberg et al., 2013* | | Transgenic |
| Antibody | anti-Brachyury (Goat polyclonal) | R and D Systems | AF2085 RRID:AB_2200235 | IF (1:200) |
| Antibody | Anti-Sox2 (Rabbit monoclonal) | Abcam | ab92494 RRID:AB_10585428 | IF (1:200) |
| Antibody | anti-Cdh1 (Goat polyclonal) | R and D Systems | AF648 RRID:AB_355504 | IF (1:200) |
| Antibody | anti-Cdh2 (Rabbit polyclonal) | Abcam | ab18203 RRID:AB_444317 | IF (1:200) |
| Antibody | anti-Tbx6 (Goat polyclonal) | R and D Systems | AF4744 RRID:AB_2200834 | IF (1:200) |
| Antibody | anti-Laminin 111 (Rabbit polyclonal) | Sigma | L9393 RRID:AB_477163 | IF (1:200) |
| Antibody | EpCAM/TROP1 (Goat polyclonal) | R and D Systems | AF960 RRID:AB_355745 | IF (1:200) |
| Antibody | anti-Vimentin (Rabbit monoclonal) | Abcam, | ab92547 RRID:AB_10562134 | IF (1:200) |
| Antibody | anti-goat 488 (Donkey polyclonal) | Molecular Probes | A11055 RRID:AB_2534102 | IF (1:1000) |
| Antibody | anti-rabbit 568 (Donkey polyclonal) | ThermoFisher Scientific | A10042 RRID:AB_2534017 | IF (1:1000) |
| Recombinant DNA reagent | *T-str* promoter | *Clements et al., 1996* | | Primitive streak-specific promoter of *T* (*Brachyury*) |
| Recombinant DNA reagent | *Snai1* cDNA | *Nieto et al., 1992* | | |
| Sequenced-based reagent | Oligonucleotides | This paper | | *Table 1* |
| Commercial assay, kit | Nextera XT index kit v2 Set B | Illumina | FC-131–2002 | |
| Commercial assay, kit | Library Quant Kit | Illumina | LC480 | |

*Continued on next page*

*Continued*

| Reagent type (species) or resource | Designation | Source or reference | Identifiers | Additional information |
|---|---|---|---|---|
| Commercial assay, kit | KAPA Library Quantification Kits | KAPA Biosystems | KK4854 | |
| Software, algorithm | SC3 | *Kiselev et al., 2017* | RRID:SCR_015953 | |
| Software, algorithm | SPRING | *Weinreb et al., 2018* | | |

## Mouse lines

*Meox2-Cre*$^{+/0}$ (*Tallquist and Soriano, 2000*) and *Snai1*$^{flox/flox}$ (*Murray and Gridley, 2006*) mouse strains were obtained from Jackson labs (B6.129S-*Snai1*$^{tm2Grid}$/J – Stock No 010686; B6.1239S4-*Meox2*$^{tm1(cre)Sor}$/J – Stock No 003755). The LuVeLu reporter strain (*Aulehla et al., 2008*) was provided by Alexander Aulehla and Olivier Pourquié. *Snai1-cKO* embryos (*Meox2-Cre*$^{+/0}$::*Snai1*$^{flox/-}$) were obtained by crossing *Meox2-Cre*$^{+/0}$::*Snai1*$^{+/-}$ males with *Snai1*$^{flox/flox}$ females. *Meox2-Cre*$^{+/0}$:: *Snai1*$^{flox/-}$::LuVeLu$^{+/0}$ embryos were obtained from similar crosses but with the LuVeLu reporter introduced into *Snai1*$^{flox/flox}$ genotype.

To obtain *T-str-Snai1* transgenic embryos, constructs were prepared by cloning a ~ 1,5 kb *Snai1* cDNA (*Nieto et al., 1992*) under the control of the PS-specific promoter of *T (Brachyury)* (*Clements et al., 1996*). These constructs were liberated from vector sequences, gel purified and used to produce transgenic embryos by pronuclear injection in FVB/N fertilized oocytes according to standard procedures (*Hogan et al., 1994*).

The *Tgfbr1*$^{+/-}$ line was generated by CRISPR/Cas9, inserting the TGATGATAGGATCC sequence, containing three stop codons and a *BamHI* restriction site in frame with the open reading frame in exon 2. For this, a gRNA containing the targeting sequence TTGACCTAATTCCTCGAGAC was produced by in vitro transcription with the T7 promoter from a plasmid derived from the gRNA-basic (*Casaca et al., 2016*). The purified gRNA was microinjected into fertilized FVB/N mouse oocytes together with the Cas9 mRNA and the synthetic ssDNA 5'-ACCACAGACAAAGTTATACACAATAG TATGTGTATAGCTGAAATTGACCTAATTCCTCGATGATGATAGGATCCGACAGGCCATTTGTATG TGCACCATCTTCAAAAACAGGGGCAGTTACTACAACATATTGC-3', containing the stop codons flanked by 60 nucleotide-long homology arms. Genotyping of embryos was performed by PCR, on DNA obtained from yolk sacs or tail biopsies from embryos or mice, respectively, as previously described (*Aires et al., 2019*). The primers used for genotyping are specified in *Table 1*.

## Ethical statement

Experiments involving animals carried out in the Oeiras laboratory followed the Portuguese (Portaria 1005/92) and European (Directive 2010/63/EU) legislations, concerning housing, husbandry, and welfare. The project was reviewed and approved by the Ethics Committee of 'Instituto Gulbenkian de Ciência' and by the Portuguese National Entity, 'Direcção Geral de Alimentação Veterinária' (license reference: 014308).

## Single cell isolation

Wild type outbred MF1 mice were crossed to obtain early head fold embryos. Caudal lateral epiblast (CLE, comprising a region immediately lateral to the posterior edge of the node and lateral to the primitive streak, extending to about half the length of the primitive streak) (*Wymeersch et al., 2016*) was microdissected in M2 medium (Sigma, M7167), using hand-pulled solid glass needles. After removal of the majority underlying paraxial mesoderm, the different tissue pieces were pooled and after 5 min at 37°C in 0.05% trypsin/EDTA, they were placed in neutralization solution [10% fetal calf serum in phosphate-buffered saline (PBS)] and dissociated by pipetting into single cells. Single cells were then transferred to a tube containing 2% FCS in PBS, strained through the 35 µm mesh of a FACS tube (Corning, 352235) and 1 µg/ml of 4',6-Diamidino-2-Phenylindole, Dihydrochloride (DAPI; Thermo Fisher Scientific, D1306) was added. Single-cells were sorted by Fluorescence-activated cell sorting FACS) with a FACS Aria II (BD Biosciences) into individual wells of a 96-well PCR plate (BioRad, HSS9601) containing the ERCC RNA spike-in Mix (Ambion, #4456740) diluted (1:100.000) in a solution of 0,2% Triton-X100 (vol/vol) (Sigma, T9284), containing 2 U/ µl of RNase

**Table 1.** Primers used for genotyping.

| | |
|---|---|
| *Snai1* deletion Fwd | CGGGCTTAGGTGTTTTCAGAC |
| *Snai1* deletion Rev | TGAAAGCGGCTCTGTTCAGT |
| *Snai1*<sup>flox</sup> Fwd | TGAAAGCGGCTCTGTTCAGTG |
| *Snai1*<sup>flox</sup> Rev | CTGCTGCACCCCTACTATGTG |
| *Meox2*-Cre Fwd | CGAGTGATGAGGTTCGCAAG |
| *Meox2*-Cre Rev | CCTGATCCTGGCAATTTCGGCT |
| LuVeLu Fwd | TGCTGCTGCCCGACAACCACT |
| LuVeLu Rev | CTTGTACAGCTCGTCCATGCC |
| *Snai1* transgenics Fwd | TTGTGTCTGCACGACCTGTGG |
| *Snai1* transgenics Rev | ATGGGGAGGTAGCAGGGTCAG |
| *Tgfbr1* Fwd | TGTGAGACAGATGGTCTTTGC |
| *Tgfbr1* mutant allele Rev | GGCCTGTCGGATCCTATCATC |
| *Tgfbr1* WT allele Rev | ACATACAAATGGCCTGTCTCG |

inhibitor (Takara Bio Europe, #2313A), 25% (v/v) of 100 μM oligo dT$_{30}$VN (Biomers) and 25% (v/v) of dNTPs 10 mM (Invitrogen, #18427013) into RNAse-free H$_2$0 (Thermo Fisher Scientific, 11430615). Plates were sealed with Microseal F (Biorad, MSF1001), centrifuged at 4°C for 1 min at 2000 rpm and stored at −80°C.

## Single-cell RNA sequencing

Full-length RNA-seq from sorted single cells was done using the Smart-seq2 method (*Picelli et al., 2014*). Libraries from a total of 91 single cells were prepared using a Nextera XT index kit v2 Set B (Illumina) and quantified both using Library Quant Kit (Illumina - LC480, KAPA Biosystems - KK4854) and the AATI Fragment Analyzer. Sequencing was done at the IGC Genomics facility (Illumina, Next-seq 500) at 5 million single end 75 bp reads per cell.

## Single-cell RNA-seq analysis

RNA-seq analysis was done using R software. Sequences were submitted to the GEO repository, accession number GSE147100. Raw sequences were aligned to the GRCm38 (mm10) reference genome using Hisat2 (*Kim et al., 2013*). The 'featureCounts' R function was used to count reads mapping to annotated genes. Clustering analysis was done using the single-cell consensus clustering (SC3) pipeline (*Kiselev et al., 2017*). Gene markers were obtained considering p-value<0,05. P-values are shown in Spreadsheet 1. Single-cell data visualization was done using SPRING (*Weinreb et al., 2018*), with the following parameters: Minimum UMI total (for filtering cells)=1000; Minimum number of cells with >= 3 counts (for filtering genes) was set to 3; Gene variability percentile (for filtering genes)=50; Number of PCA dimensions (for building graph) was set to 20; Number of nearest neighbours (for graph)=5. Single cell heatmaps were created using the Heatmapper platform (*Babicki et al., 2016*). The scRNA-seq values in RPKM used for this analysis are shown Spreadsheet 2.

## In Situ hybridization and sectioning

Wholemount in situ hybridization was performed as previously described (*Aires et al., 2019*). These experiments were repeated independently at least twice for each genotype giving identical results, with the exception of *Lfng*, which gave different patterns in the PSM region as reported in *Figure 5* and *Figure 5—figure supplement 1*, and notochord markers in *Snai1-cKO* embryos, that gave two alternative patterns as referenced in the main text. Post-stained embryos were included in a mixture of 0.45% gelatin (Merck), 27% bovine serum albumin (Roche), 18% sucrose (Sigma) in PBS that was then jellified with 1.75% glutaraldehyde (Biochem chemopharma) and sectioned at 20 μm with a vibratome (Leica).

## Immunohistochemistry and 3D imaging

Wholemount immunofluorescence staining of tail tissues was performed as previously described (*Osorno et al., 2012*). At least two embryos per genotype were stained with each antibody with similar results. Primary antibodies (1:200): goat anti-Brachyury (R and D Systems, AF2085), rabbit anti-Sox2 (Abcam, AB92494), goat anti-Cdh1 (R and D Systems, AF648), rabbit anti-Cdh2 (Abcam, AB18203), goat anti-Tbx6 (R and D Systems, AF4744), rabbit anti-Laminin 111 which detects all laminins containing a1, b1 or g1 chains (Sigma, L9393), goat anti-human EpCAM/TROP1 (R and D Systems, AF960) and rabbit anti-Vimentin (Abcam, AB92547). Secondary antibodies (1:1000): donkey anti-goat 488 (Molecular Probes, A-11055) and donkey anti-rabbit 568 (Molecular Probes, A10042). Immuno-stained tails were imaged on a Prairie two-photon system, using an Olympus 20 × 1.0 NA W objective, with the excitation laser tuned to 960 nm, and GaAsP photodetectors. Z stacks of 1024 × 1024 images were acquired every 1 µm, with either 1x or 1.5x zoom. Laser intensity and photomultiplier levels were maintained across replicates and controls.

## Live imaging (LuVeLu)

Embryos expressing the LuVeLu reporter (*Aulehla et al., 2008*) were dissected in pre-warmed (at 37°C) M2 medium (Sigma) and cultured in low glucose DMEM medium (Gibco, 11054020), 10% of HyClone defined fetal bovine serum (GE Healthcare, #HYCLSH30070.03), 2 mM of L-glutamine (Gibco,#25030–024) and 1% penicillin-streptomycin (Sigma, #P0781). Embryos were cultured at 37°C in a 65% $O_2$ and 15% $CO_2$ environment (N2 balanced). Embryos were imaged on the Prairie two-photon system (laser tuned to 960 nm). At time = 0, a z-stack of 5 µm step-size was acquired at 1024 × 1024 pixel size, using the 20x objective. From t = 1 and onwards, we acquired z-stacks series of 1024 × 512 pixel images, spaced in depth at 10 µm, using a Nikon 16x LWD 0.8NA W objective. T-series were acquired every 8,5 min.

## Image processing

3D and 4D series of two-photon microscopy datasets were processed using Fiji (*Schindelin et al., 2012*). Pre-processing involved removal of outlier pixels and elimination of electronic noise in the form of periodic patterns by using fast-Fourier filters (detection of 'maxima' in the Fourier spectrum and deleting for each an area with r = 5 pixels, performed slice-by-slice). 3D datasets of immuno-stained tissues were deconvolved using Huygens (SVI). When acquired, adjacent 3-stack datasets were digitally stitched using the 'Image Stitching' plugin (*Preibisch et al., 2009*) (pairwise mode, with alpha blending). We then evaluated the Z-axis attenuation in 3D datasets; when attenuation in depth was noticeable we performed a compensation using FIJI's 'Math...Batch' function and the expression: v = v * exp (a * z), where v = pixel intensity; a = compensation factor (ranging from 1 to 2, depending on the compensation necessary to equalize the intensities on the first and the deeper optical slices) – replicates and controls were treated similarly. The pre-processed multi-channel dataset was then repositioned by affine transformation using the 'TransformJ' plugin (*Meijering et al., 2001*). The transformation matrix was obtained using FIJI's 3D viewer plugin. 3D visualization and analysis of embryo tails was done using Imaris v9.0 (Bitplane), including the rendering of wholemount (blend mode) and of sagittal and transversal 5 µm optical sections by maximum intensity projection. Note: In EPCAM-stained samples, a considerable bleed-through from the DAPI channel was observed. Therefore, before deconvolution, the EPCAM channel was compensated by dividing it by the DAPI signal. Further background reduction was achieved by dividing the compensated dataset by a 'synthetic background' obtained from optical slices without stained tissue. Mutant and wild type embryos were treated similarly.

A CLAHE filter was used to enhance the contrast of DAPI-stained embryos which were also imaged using the Prairie two-photon system (20x objective and laser tuned to 890 nm). Defined embryonic structures (e.g. neural tube) were then segmented by manual contouring and 3D rendered using Amira (Thermo-Fisher Scientific).

To analyse the LuVeLu time-series, we concatenated all 3D datasets and converted to a 4D dataset in HDF5 format. The time-points were then registered using the BigStitcher plugin (*Hörl et al., 2019*). Visualization and analysis were done using Imaris.

## Acknowledgements

We would like to thank Achim Gossler for the *T-Str* enhancer, Olivier Pourquié and Alexander Aulehla for the LuVeLu reporter strain, Sólveig Thorsteinsdóttir for the laminin antibody, Angela Nieto for *Snai1* cDNA, Julien Delile for sharing data files, Luisa de Lemos for sharing unpublished data, Daniel Neves for the help with the analysis of the RNA-seq data, Hugo Pereira for the help using BigStitcher and Nuno Granjeiro for helping to set up the live imaging apparatus. We also would like to thank the IGC and SCRM animal facilities, Fiona Rossi and Clair Cryer from the SCRM flow cytometry facility, João Sobral from the IGC genomics facility, and past and present members of the Mallo lab for useful comments and support during the course of this project. This work has been supported by grants PTDC/BEX-BID/0899/2014 and LISBOA-01–0145-FEDER-030254 (FCT, Portugal) and SCML-MC-60–2014 (Santa Casa da Misericórdia, Portugal) to M.M.; the research infrastructure Congento, project LISBOA-01–0145-FEDER-022170; PhD fellowships PD/BD/128426/2017 to AD and (PD/BD/128437/2017) to AL (FCT, Portugal); MRC (UK) grants MR/S008799/1 and MR/K011200/1 to V.W. and a Development Travelling Fellowship (DEV-170806) from 'The Company of Biologists' to AD.

## Additional information

### Funding

| Funder | Grant reference number | Author |
|---|---|---|
| Fundação para a Ciência e a Tecnologia | PTDC/BEX-BID/0899/2014 | Moises Mallo |
| Fundação para a Ciência e a Tecnologia | LISBOA-01-0145-FEDER-030254 | Moises Mallo |
| Fundação para a Ciência e a Tecnologia | LISBOA-01-0145-FEDER-022170 | Moises Mallo |
| Fundação para a Ciência e a Tecnologia | PD/BD/128426/2017 | André Dias |
| Fundação para a Ciência e a Tecnologia | PD/BD/128437/2017 | Anastasiia Lozovska |
| Santa Casa da Misericórdia | SCML-MC-60-2014 | Moises Mallo |
| Medical Research Council | MR/S008799/1 | Valerie Wilson |
| Medical Research Council | MR/K011200/1 | Valerie Wilson |
| Company of Biologists | DEV-170806 | André Dias |

The funders had no role in study design, data collection and interpretation, or the decision to submit the work for publication.

### Author contributions

André Dias, Conceptualization, Formal analysis, Investigation, Writing - review and editing; Anastasiia Lozovska, Ana Nóvoa, Investigation; Filip J Wymeersch, Anahi Binagui-Casas, Investigation, Methodology; Daniel Sobral, Formal analysis, Methodology; Gabriel G Martins, Data curation, Methodology; Valerie Wilson, Supervision, Funding acquisition, Methodology; Moises Mallo, Conceptualization, Supervision, Funding acquisition, Investigation, Writing - original draft, Project administration, Writing - review and editing

### Author ORCIDs

André Dias ![ORCID] https://orcid.org/0000-0003-3337-6373
Anastasiia Lozovska ![ORCID] https://orcid.org/0000-0002-9842-6450
Filip J Wymeersch ![ORCID] https://orcid.org/0000-0001-8999-4555
Ana Nóvoa ![ORCID] https://orcid.org/0000-0002-5668-5630
Anahi Binagui-Casas ![ORCID] https://orcid.org/0000-0002-7987-9286
Daniel Sobral ![ORCID] https://orcid.org/0000-0003-3955-0117

Gabriel G Martins (iD) http://orcid.org/0000-0002-6506-9776
Valerie Wilson (iD) https://orcid.org/0000-0003-4182-5159
Moises Mallo (iD) https://orcid.org/0000-0002-9744-0912

## Ethics

Animal experimentation: Experiments involving animals carried out in the Oeiras laboratory followed the Portuguese (Portaria 1005/92) and European (Directive 2010/63/EU) legislations, concerning housing, husbandry, and welfare. The project was reviewed and approved by the Ethics Committee of "Instituto Gulbenkian de Ciência" and by the Portuguese National Entity, "Direcção Geral de Alimentação Veterinária" (license reference: 014308).

## Decision letter and Author response

Decision letter https://doi.org/10.7554/eLife.56615.sa1
Author response https://doi.org/10.7554/eLife.56615.sa2

# Additional files

## Supplementary files

- Transparent reporting form

## Data availability

Sequencing data have been deposited in GEO under accession code GSE147100.

The following dataset was generated:

| Author(s) | Year | Dataset title | Dataset URL | Database and Identifier |
|-----------|------|---------------|-------------|-------------------------|
| Dias A, Mallo M | 2020 | Single-cell RNA sequencing of neuromesodermal progenitors of early headfold wild type embryos | https://www.ncbi.nlm.nih.gov/geo/query/acc.cgi?acc=GSE147100 | NCBI Gene Expression Omnibus, GSE147100 |

The following previously published datasets were used:

| Author(s) | Year | Dataset title | Dataset URL | Database and Identifier |
|-----------|------|---------------|-------------|-------------------------|
| Aires R, Mallo M | 2018 | RNA-Seq of mus musculus: Tailbud WT1 | https://www.ncbi.nlm.nih.gov/sra/SRX4968732 | NCBI Sequence Read Archive, SRX4968732 |
| Aires R, Mallo M | 2018 | RNA-Seq of mus musculus: Tailbud WT2 | https://www.ncbi.nlm.nih.gov/sra/SRX4968731 | NCBI Sequence Read Archive, SRX4968731 |
| deLemos L, Mallo M | 2019 | RNA-seq of mus musculus : Tail Bud progenitors 2 | https://www.ncbi.nlm.nih.gov/sra/SRX5532193 | NCBI Sequence Read Archive, SRX5532193 |
| deLemos L, Mallo M | 2019 | RNA-seq of mus musculus : Tail Bud progenitors 1 | https://www.ncbi.nlm.nih.gov/sra/SRX5532192 | NCBI Sequence Read Archive, SRX5532192 |
| Wymeersch FJ, Skylaki S, Huang Y, Watson JA, Economou C, Marek-Johnston C, Tomlinson SR, Wilson V | 2018 | Gene expression in microdissected embryonic regions during mouse axis elongation. | https://www.ncbi.nlm.nih.gov/geo/query/acc.cgi?acc=GSE120870 | NCBI Gene Expression Omnibus, GSE120870 |
| Briscoe J, Kleinjung J, Delile J, Gouti M | 2015 | Retinoic acid mediated neural and mesoderm specification during vertebrate trunk | https://www.ebi.ac.uk/arrayexpress/experiments/E-MTAB-5208 | ArrayExpress, E-MTAB-5208 |

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
