## [Decision Letter]

**Acceptance summary:**

This study identifies the critical role played by Tgfb1 and Snail in formation and maintenance of the so-termed "tail bud", a transient population present in the posterior region of the post-implantation embryo that is essential for generating the secondary axial structures. Importantly Tgfb1 and Snail activities within the tail bud maintain the progenitor cell population necessary to drive axial extension.

**Decision letter after peer review:**

Thank you for submitting your article "A TgfbRI/Snai1-dependent developmental module at the core of vertebrate axial elongation" for consideration by *eLife*. Your article has been reviewed by two peer reviewers, and the evaluation has been overseen by a Reviewing Editor and Kathryn Cheah as the Senior Editor. The following individuals involved in review of your submission have agreed to reveal their identity: Thomas Gridley (Reviewer #1); Ben Steventon (Reviewer #3).

The reviewers have discussed the reviews with one another and the Reviewing Editor has drafted this decision to help you prepare a revised submission.

Summary:

The manuscript presents a comprehensive analysis of the independent regulation of the generation of the primary (anterior to the sacrum; primitive streak-derived) versus the secondary (posterior to the sacrum; tail bud-derived) body of the mammalian embryo, using the mouse as a model system.

Revisions:

The use of the LuVeLu transgenic reporter would benefit from a more thorough description of what it actually consists of, and what type of data it is supplying. A similar point can be made about the T-str enhancer, and its use for *Snai1* overexpression.

The term “developmental module” is introduced in the Abstract, but it is not defined or mentioned anywhere else in the manuscript. Please either remove this term, or provide additional explanation in the Introduction and/or Discussion.

The statement "the vertebrate body is laid down progressively in a head to tail sequence by dedicated axial progenitors with stem-cell like properties" needs to be revised. While this may be true for mouse embryos, it is not clear at all that this description is true for all vertebrates. In *Xenopus*, for example, much of the vertebrate body axis is established during gastrulation by inductive mechanisms in response to the organiser. The idea that continued addition of cells from posterior progenitor populations is a conserved mechanism for building the vertebrate body axis is misleading. The subsequent sentences should also be clear that this refers for evidence from mouse embryos.

Describing the tailbud as a “blastema-like structure” is misleading, as there is little evidence supporting this claim.

Throughout the manuscript there is little indication as to the number of embryos that have been examined for each result presented. Please include this information the revised version of the manuscript.

The expression pattern of *Snai1* should be described prior to discussing the mutant phenotype. The same is required when introducing experiments related to the *Alk5* mutants. It would also be useful to point out where the expression of these genes are observed in the scRNAseq datasets.

---

## [Author Response]

Revisions:The use of the LuVeLu transgenic reporter would benefit from a more thorough description of what it actually consists of, and what type of data it is supplying. A similar point can be made about the T-str enhancer, and its use for Snai1 overexpression.

We have now introduced changes to better clarify the use of the LuVeLu and the T-str enhancer. For the LuVeLu this includes explanatory text (subsection “*Snai1* is required for axial progenitor mobilization to form the tail bud” paragraph three) and a supporting supplementary figure (Figure 5—figure supplement 1) showing in situ hybridization images of Lfng expression in Snai1-cKO mutant embryos that prompted live image analyses of Lfng cycling activity using the LuVeLu reporter. In the case of the T-str, a short explanation is provided in subsection “Extended *Snai1* expression in the PS mobilizes functional tail bud progenitors”.

The term “developmental module” is introduced in the Abstract, but it is not defined or mentioned anywhere else in the manuscript. Please either remove this term, or provide additional explanation in the Introduction and/or Discussion.

We have introduced modifications in the text to address this issue. These include the last part of the Introduction. We already had a note about it in the Discussion (paragraph three).

The statement "the vertebrate body is laid down progressively in a head to tail sequence by dedicated axial progenitors with stem-cell like properties" needs to be revised. While this may be true for mouse embryos, it is not clear at all that this description is true for all vertebrates. In *Xenopus,* for example, much of the vertebrate body axis is established during gastrulation by inductive mechanisms in response to the organiser. The idea that continued addition of cells from posterior progenitor populations is a conserved mechanism for building the vertebrate body axis is misleading. The subsequent sentences should also be clear that this refers for evidence from mouse embryos.

This is actually a very important comment for the sake of accuracy in our manuscript. We have changed the initial sentences of the Introduction to address this comment.

Describing the tailbud as a “blastema-like structure” is misleading, as there is little evidence supporting this claim.

We just removed the description.

Throughout the manuscript there is little indication as to the number of embryos that have been examined for each result presented. Please include this information the revised version of the manuscript.

We have added the numbers in the relevant places.

The expression pattern of Snai1 should be described prior to discussing the mutant phenotype. The same is required when introducing experiments related to the Alk5 mutants. It would also be useful to point out where the expression of these genes are observed in the scRNAseq datasets.

We have now described the *Snai1* expression pattern at the beginning of the Snai1-related section (subsection “*Snai1* is required for axial progenitor mobilization to form the tail bud”) and for *Alk5* in subsection “*Snai1* and *Tgfbr1* cooperatively orchestrate the transition from primary to secondary body formation”. Also, we had included two supplementary figures (Figure 2—figure supplement 1 and 2) describing the expression of *Snai1* and *Alk5* in the scRNAseq datasets.